# Bootstrapping the error of Oja's algorithm

**Robert Lunde**
University of Michigan
rlunde@umich.edu

**Purnamrita Sarkar**
University of Texas at Austin
purna.sarkar@austin.utexas.edu

**Rachel Ward**
University of Texas at Austin
rward@math.utexas.edu

## Abstract

We consider the problem of quantifying uncertainty for the estimation error of the leading eigenvector from Oja's algorithm for streaming principal component analysis, where the data are generated IID from some unknown distribution. By combining classical tools from the U-statistics literature with recent results on high-dimensional central limit theorems for quadratic forms of random vectors and concentration of matrix products, we establish a weighted $\chi^2$ approximation result for the $\sin^2$ error between the population eigenvector and the output of Ojas algorithm. Since estimating the covariance matrix associated with the approximating distribution requires knowledge of unknown model parameters, we propose a multiplier bootstrap algorithm that may be updated in an online manner. We establish conditions under which the bootstrap distribution is close to the corresponding sampling distribution with high probability, thereby establishing the bootstrap as a consistent inferential method in an appropriate asymptotic regime.

## 1 Introduction

Since its discovery over a century ago [13], principal component analysis (PCA) has been a cornerstone of data analysis. In many applications, dimension reduction is paramount and PCA offers an optimal low-rank approximation of the original data. PCA is also highly interpretable as it projects the dataset onto the directions that capture the most variance known as principal components.

Important applications of PCA include image and document analysis, where the largest few principal components may be used to compress a large dimensional dataset to a manageable size without incurring much loss; for a discussion of some other applications of PCA, see for example, [28]. In these settings, the original dimensionality, which could be the number of pixels in an image or the vocabulary size after removing stop-words, is in the tens of thousands. An offline computation of the principal components would require the computation of eigenvectors of the sample covariance matrix. However, in high-dimensional settings, storing the covariance matrix and subsequent eigen-analysis can be challenging. Streaming PCA methods have gained significant traction owing to their ability to iteratively update the principal components by considering one data-point at a time.

One of the most widely used algorithms for streaming PCA is Oja's algorithm, proposed in the seminal work of [41]. Oja's algorithm involves the following update rule:

$$w_{t+1} - w_t = \eta(w_t^T X_t) X_t; \qquad w_{t+1}^T w_{t+1} = 1, \tag{1}$$

where $X_t \in \mathbb{R}^d$ is the $t^{th}$ data point and $w_t$ is the current estimate for the leading eigenvector of $\Sigma = \mathbb{E} X X^T$ after $t$ data-points have been seen. The parameter $\eta$ can be thought of as a learning rate, which can either be fixed or varied as a function of $t$. In this paper we fix the learning rate, similar to [26].

35th Conference on Neural Information Processing Systems (NeurIPS 2021).

**Contribution:** In the present work, we consider the problem of uncertainty quantification for the estimation error of the leading eigenvector from Oja's algorithm, which is one of the most commonly used streaming PCA algorithms. Our contributions may be summarized as follows:

1. We derive a high-dimensional weighted $\chi^2$ approximation to the $\sin^2$ error for the leading eigenvector of Oja's algorithm. We recover the optimal convergence rate $O(1/n)$ while allowing $d$ to grow at a sub-exponential rate under suitable structural assumptions on the covariance matrix, matching state-of-the-art theoretical results for consistency of Oja's algorithm. Our result provides a distributional characterization of the $\sin^2$ error for Oja's algorithm for the first time in the literature. The approximation holds for a wide range of step sizes.

2. Since the weighted $\chi^2$ approximation depends on unknown parameters, we propose an online bootstrap algorithm and establish conditions under which the bootstrap is consistent. Our bootstrap procedure allows the approximation of important quantities such as the quantiles of the error associated with Oja's algorithm for the first time.

**Prior analysis of Oja's algorithm.** While Oja's algorithm was invented in 1982 it was not until recently that the theoretical workings of Oja's algorithm have been understood. A number of papers in recent years have focused on proving guarantees of convergence of the iterative update in (1) toward the principal eigenvector of the (unknown) covariance matrix $\mathbb{E}XX^T$, which can be recast as stochastic gradient descent (SGD) on the quadratic objective function

$$\min_{\substack{w \\ w^T w = 1}} -\text{trace}(w^T \Sigma w), \qquad \Sigma = \mathbb{E}XX^T, \tag{2}$$

projected onto the non-convex unit sphere. We assume that the data-points are mean zero. Despite being non-convex and thus falling outside the framework for which theory for stochastic gradient descent convergence is firmly established, the output of Oja's algorithm be viewed as a product of random matrices and shares similar structure to other important classes of non-convex problems, such as matrix completion [27, 29], matrix sensing [27], and subspace tracking [4]. Thus, studying this optimization problem serves as a natural first step toward understanding the behavior of SGD in more general non-convex settings.

Let $v_1$ denote the principal eigenvector of $\Sigma$, and let $\hat{v}_1 = w_n$ be the solution to the stochastic iterative method applying Eq 1. Finally, let $\lambda_1 > \lambda_2$ be the first and second principal eigenvalues of $\Sigma$. Sharp rates of convergence for Oja's updates were established in [25]. Under boundedness assumptions on $\|X_i X_i^T - \Sigma\|$, they show that with constant probability, the square of the sine of the angle between $v_1$ and $w$ satisfies:

$$1 - (v_1^T \hat{v}_1)^2 = O\left(\frac{1}{n}\right) \tag{3}$$

where the $O$ hides a constant which depends in the optimal way on the eigengap between the top two eigenvalues, and independent of $n$ or $d$, improving on previous error bounds for Oja's algorithm [46, 18, 3, 47, 38, 2] which showed convergence rates that deteriorate with the ambient dimension $d$, and thus did not fully explain the efficiency of Oja's update. This sharp rate is remarkable, as it matches the error of the principal eigenvector of the sample covariance matrix, which is the batch or offline version of PCA. Other notable work include [31, 33] for unbounded $X_i$, analysis of Oja's algorithm for computing top $k$ principal components [1, 24].

**The bootstrap.** The bootstrap, proposed by [9], is one of the most widely used methods for uncertainty quantification in machine learning and statistics and accordingly has a vast literature. We refer the reader to [17, 49] for expositions on the classical theory of the bootstrap for IID data. Recently, since the groundbreaking work of [7, 8], the bootstrap has seen a renewed surge of interest in the context of high-dimensional data where $d$ can be potentially exponentially larger than $n$. Of particular relevance to the present work are high-dimensional central limit theorems (CLTs) for quadratic forms, which have been studied by [43, 51, 15]. In particular, our CLT for the estimation error of Oja's algorithm invokes a modest adaptation of [51] to independent but non-identically distributed random variables. In machine learning, bootstrap methods have been used to estimate the uncertainty of randomized algorithms such as bagging and random forests [35], sketching for large scale singular value decomposition (SVD) [36], randomized matrix multiplication [37], and randomized least squares [34].

A standard notion of bootstrap consistency is that, conditioned on the data, the distribution of the suitably centered and scaled bootstrap functional approaches the true distribution with high probability in some norm on probability measures, typically the Kolmogorov distance, which is the supremum of the absolute pointwise difference between two CDFs. Bootstrap consistency is often established by deriving a Gaussian approximation for the sampling distribution and showing that the bootstrap distribution is close to the corresponding Gaussian approximation with high probability.

It may seem that if one knows that the approximating distribution of a statistic is Gaussian, this defeats the purpose of bootstrap. However, *for most statistics, the parameters of the normal approximation depend on unknown model parameters, and have to be estimated if one intends to use the normal approximation*. Furthermore, the CLT only gives a *first-order* correct approximation of the target distribution, i.e. with $O(1/\sqrt{n})$ error. In contrast, the bootstrap of a suitably centered and scaled statistic has been shown to be higher order correct for many functionals [16, 17, 19].

**Quantifying uncertainty for SGD.** Behind the recent success of neural networks in a wide range of sub-fields of machine learning, the workhorse algorithm has become Stochastic gradient descent (SGD) [42, 40, 44]. For establishing consistency of bootstrap, one requires to establish asymptotic normality [11, 42, 45, 39]. There has also been many works on uncertainty estimation of SGD [6, 32, 12, 48]. However, *all these works are for convex, and predominantly strongly convex loss functions*. Only recently, [52] has established asymptotic normality for nonconvex loss functions under dissipativity conditions and appropriate growth conditions on the gradient, which are weaker conditions than strong convexity but not significantly so.

Now, in Section 2 we present notation and do setup, present our main theoretical results in Section 3, followed by simulations in Section 4.

## 2  Preliminaries

We consider a row-wise IID triangular array, where the random vectors $\{X_i\}$ in the $n^{th}$ row take values in $\mathbb{R}^{d_n}$, with $\mathbb{E}[X_i] = 0$ and $\mathrm{Var}(X_i) = \Sigma_n$. Note that the triangular array allows $\{X_1, \ldots, X_n\}$ come from a different distribution for each $n$ and the setting where $d$ is fixed and $n$ grows is a special case. For readability, we drop the subscript $n$ from $\Sigma_n$. We use $\|\cdot\|$ to denote the Euclidean norm for vectors and the operator norm for matrices and $\|\cdot\|_F$ to denote the Frobenius norm.

Expanding out the recursive definition in Eq 1, we see that Oja's iteration can be expressed as $w_{t+1} = (I_d + \eta X_t X_t^T)w_t$. Thus, after $n$ iterations the vector can be written as a matrix-vector product, where the matrix is a product of $n$ independent matrices. Expanding out the recursive definition, we get:

$$B_n := \prod_{i=1}^{n}(I_d + \eta X_i X_i^T) \qquad \hat{v}_1 = \frac{B_n u_0}{\|B_n u_0\|}, \tag{4}$$

where $I_d$ is a $d \times d$ identity matrix. where $u_0$ is a random unit vector in $d$ dimensions. In the scalar case, when $\eta = 1/n$, for large $n$, the numerator of Eq 4 behaves like $\exp(\sum_i X_i^2/n)$, which in turn converges to $\exp(E[X_1^2])$. For matrices, one hopes that, by independence, a result of the same flavor will hold. And in fact if it does hold, then for $\eta = \frac{\log n}{n}$, the numerator in Eq 4 will concentrate around $\exp(\log n \Sigma)$. The spectrum of this matrix is dominated by the principal eigenvector, i.e. the ratio of the first eigenvalue to the second one is $\exp(\log n(\lambda_1 - \lambda_2))$, where $\lambda_i$ is the $i^{th}$ eigenvalue of the covariance matrix $\Sigma$. This makes it clear that *Oja's algorithm is essentially a matrix vector product of this matrix exponential (suitably scaled) and a random unit vector.*

However, the intuition from the scalar case is nontrivial to generalize to matrices due to non-commutativity. Limits of products of random matrices have been studied in mathematics in the context of ergodic theory on Markov chains (see [14, 30, 5, 10] etc.). However, until recent results of [23], which extended and improved results in [21], there has not been much work on quantifying the exact *rate of convergence*, or finite-sample large deviation bounds for how a random matrix product deviates from its expectation.

We reparametrize $\eta$ as $\eta_n/n$, where $\eta_n$ is chosen carefully to obtain a suitable error rate. Note that this is not a scheme where we decrease $\eta$ over time as in [20], but hold it as a constant which is a function of the total number of data-points.

## 2.1 The Hoeffding decomposition

The Hoeffding decomposition, attributed to [22], is a key technical tool for studying the asymptotic properties of U-statistics. However, the idea generalizes far beyond U-statistics; see Supplement Section A for further discussion. In the present work, we use Hoeffding decompositions for matrix and vector-valued functions of independent random variables taking values in $\mathbb{R}^d$ to facilitate analysis for $B_n$.

A concept closely related to the Hoeffding decomposition is the more well-known Hájek projection, which gives the best approximation (in an $L_2$ sense) of a general function of $n$ independent random variables by a function of the form $\sum_i g_i(X_i)$, where $g_i$ are measurable functions satisfying a square integrability condition. The Hájek projection facilitates distributional approximations for complicated statistics since this linear projection is typically more amenable to analysis. However, establishing a central limit theorem requires showing the negligibility of a remainder term, which can be large if the projection is not accurate enough.

The Hájek projection may be viewed as the first-order term in the Hoeffding decomposition, a general way of representing functions of independent random variables. The Hoeffding decomposition consists of a sum of projections onto a linear space, quadratic space, cubic space, and so on. Each new space is chosen to be orthogonal to the previous space. Thus, the Hoeffding decomposition can be thought of as a sum of terms of increasing levels of complexity. Even if the remainder of the Hájek projection turns out to be small, the Hoeffding decomposition can be easier to work with due to the orthogonality of the projections.

**The Hoeffding decomposition for the matrix product.** Let $Y_i = X_i X_i^T - \Sigma$ and let $S \subseteq \{1, \ldots n\}$. By Corollary A.1 of the Supplement Section A, the Hoeffding Decomposition for $B_n$ is given by:

$$B_n = \sum_{k=0}^{n} T_k, \qquad T_k = \sum_{|S|=k} H^{(S)}. \tag{5}$$

where $H^{(S)} = \prod_{i=1}^{n} A_i^{(S)}$ and $A_i^{(S)}$ is given by: $A_i^{(S)} = \begin{cases} \frac{\eta_n}{n} Y_i & \text{if } i \in S \\ I + \frac{\eta_n}{n} \Sigma & \text{otherwise} \end{cases}$.

The above expansion has favorable properties that facilitate second-moment calculations. In fact, as a consequence of the orthogonality property of Hoeffding projections, we have that

$$\mathbb{E}\left[\|B_n\|_F^2\right] = \sum_{k=0}^{n} \sum_{|S|=k} \mathbb{E}\left[\|\prod_{i=1}^{n} H_i^{(S)}\|_F^2\right]$$

$$\mathbb{E}\left[\|B_n x\|^2\right] = \sum_{k=0}^{n} \sum_{|S|=k} \mathbb{E}\left[\|\prod_{i=1}^{n} H_i^{(S)} x\|^2\right]$$

where the second statement holds for any $x \in \mathbb{R}^d$; see Proposition A.2 in Supplement Section A.

## 2.2 Online bootstrap for streaming PCA

To approximate the sampling distribution, we consider a Gaussian multiplier bootstrap procedure. As observed by [7], a Gaussian multiplier random variable eliminates the need to establish a Gaussian approximation for the bootstrap since conditional on the data, it is already Gaussian. It is not hard to see that this is a natural candidate for the online setting; the multiplier bootstrap has been used for bootstrapping the stochastic gradient descent estimator in [12].

We present our bootstrap in Algorithm 1. In our procedure, we update $m + 1$ vectors at every iteration. The first one is $\hat{v}$, which will result in the final Oja estimate of the first principal component. The other vectors $\{v^{*(j)}, j = 1, \ldots m\}$ are obtained by perturbing the basic Oja update (Eq 1).

The $W_i$'s are the multiplier random variables, which are scaled mean zero scaled Gaussians with variance $1/2$. The update of the $v^{*(j)}$ is novel because it preserves the mean and the variance of the original Oja estimator while not requiring access to the full sample covariance matrix. Consequently, we can make our updates online and attain both a point estimate and a confidence interval for the principal eigenvector, *while increasing the computation and storage by only a factor of $m$.*

---

**Algorithm 1:** Bootstrap for Oja's algorithm

---

**Input**: Datapoints $X_1, \ldots, X_n$, stepsize $\eta$, number of bootstrap replicates $m$

**Output**: Oja's solution $\hat{v}_1$ and $m$ bootstrapped versions of it $v_1^{*(1)}, \ldots, v_1^{*(m)}$

Draw $g \sim N(0, I_d)$

Create unit vector $u_0 \leftarrow g/\|g\|$

Initialize $\hat{v}_1, v_1^{*(1)}, \ldots, v_1^{*(m)} \leftarrow u_0$

**for** $t=2,\ldots, n$ **do**

    Update $\hat{v}_1 \leftarrow \hat{v}_1 + \eta(X_t^T \hat{v}_1)\hat{v}_1$

    Normalize $\hat{v}_1$ to have unit norm;

    **for** $i=1{:}m$ **do**

        Draw $W_i \sim N(0, 1/2)$;

        Let $h^{(i)} \leftarrow (X_t^T v_1^{*(i)})X_t$;

        Let $g^{(i)} \leftarrow (X_{t-1}^T v_1^{*(i)})X_{t-1}$;

        Update $v_1^{*(i)} \leftarrow v_1^{*(i)} + \eta\left(h^{(i)} + W_i(h^{(i)} - g^{(i)})\right)$;

        Normalize $v_1^{*(i)}$ to have unit norm;

    **end**

**end**

---

## 3 Main results

In this section we present our main contributions: a CLT for the error of Oja's algorithm and consistency of an online multiplier bootstrap for error.

### 3.1 Central limit theorem for the error of Oja's algorithm

We start by stating a CLT for the error of Oja's algorithm. To state this theorem, we will need to introduce some notation.

Let $\hat{v}_1$ denote the Oja vector and $V_\perp$ the $d \times d-1$ matrix with $2, \ldots, d$ eigenvectors of $\Sigma$ on its columns. Note that $V_\perp$ is not uniquely defined, but $V_\perp V_\perp^T = I - v_1 v_1^T$ is if the leading eigenvalue is distinct and consequently, norms of the form $\|V_\perp^T x\|$ for $x \in \mathbb{R}^d$ are well-defined. Let $\lambda_1 \geq \cdots \geq \lambda_d$ denote the eigenvalues of $\Sigma$ and $\Lambda_\perp$ be a diagonal matrix with $\Lambda_\perp(i,i) = (1 + \eta_n \lambda_{i+1}/n)/(1 + \eta_n \lambda_1/n)$, $i = 1, \ldots, d-1$. Also let

$$\mathbb{M} := \mathbb{E}\left[V_\perp^T (X_1^T v_1)^2 X_1 X_1^T V_\perp\right] \tag{6}$$

Now we define

$$\bar{\mathbb{V}}_n = \frac{\eta_n}{n} \sum_i \mathbb{E}[V_\perp \Lambda_\perp^{i-1} V_\perp^T (X_i X_i^T - \Sigma) v_1 v_1^T (X_i X_i^T - \Sigma) V_\perp \Lambda_\perp^{i-1} V_\perp^T]$$

$$= \frac{\eta_n}{n} V_\perp \left(\sum_i \Lambda_\perp^{i-1} \mathbb{M} \Lambda_\perp^{i-1}\right) V_\perp^T \tag{7}$$

We have the following result:

**Theorem 1.** *Suppose that $u_0$ is drawn from the uniform distribution on $\mathcal{S}^{d-1}$, $\lambda_1 = O(1)$. Choose $\eta_n \to \infty$ such that $nd \cdot \exp(-\eta_n(\lambda_1 - \lambda_2)) \to 0$, $\frac{(\eta_n \vee \log d)\, \eta_n^2 (M_d^2 \vee 1)}{n} \to 0$, where $M_d = \mathbb{E}[\|X_i X_i^T - \Sigma\|^2]$. Further, let $\widetilde{Z}_n$ be a mean $0$ Gaussian matrix such that $Var(\widetilde{Z}_n) = Var((X_1 X_1^T - \Sigma)v_1)$ and suppose that:*

$$\|\mathbb{M}\|_F \geq c > 0 \tag{8}$$

$$\frac{\mathbb{E}\left[\left\|V_\perp^T \widetilde{Z}_n\right\|^6\right] \vee \mathbb{E}\left[\left\|V_\perp^T (X_1 X_1^T - \Sigma)v_1\right\|^6\right]}{\|\mathbb{M}\|_F^3} = o(n) \tag{9}$$

Then, for a sequence of Gaussian distributions $\{Z_n\}_{n \geq 1}$ with mean $0$ and covariance matrix $\bar{\mathbb{V}}_n$ (see Eq 7), the following holds:

$$\sup_{t \in \mathbb{R}} \left| P\left(n/\eta_n \ \sin^2(\hat{v}_1, v_1) \leq t\right) - P(Z_n^T Z_n \leq t) \right| \to 0 \tag{10}$$

Theorem 1 is very general. We allow the dimension to grow with the number of observations, which is typical in the high-dimensional bootstrap literature. Note that the case of fixed $d$ and growing $n$ is also a special case of this setup.

We want to point out that while previous literature obtained sharp bounds on the $\sin^2$ error $1 - (v_1^T \hat{v}_1)^2$, we go a step further. *We establish an approximating distribution for $n/\eta_n (1 - (v_1^T \hat{v}_1)^2)$.*

**Remark 1** (Condition on norm). *For simplicity, we assume $\lambda_1 = O(1)$, which can be easily relaxed to grow slowly with $n$. We do not assume that the $\|X_i X_i^T - \Sigma\|_2$ is bounded almost surely. However, the norm of $X_i X_i^T - \Sigma$ comes into play implicitly via the assumption in Eq 9. Consider the case where $X_i$ are drawn from some multivariate Gaussian distribution. We use this to build intuition about the assumptions in Eq 8 and 9. In this case, $X_1^T V_\perp$ is a Gaussian of independent entries and thus $\mathbb{E}\left[ \left\| V_\perp (X_1 X_1^T - \Sigma) v_1 \right\|^6 \right] = \mathbb{E}\|X_1^T v_1\|^6 \mathbb{E}\left( \sum_{j>1}(X_j^T v_j)^2 \right)^3$. Note that $\sum_{j>1}((X_j^T v_j)^2 - \lambda_j)$ is a sub-exponential random variable with parameters $(c_1 \sum_{j>1} \lambda_j, c_2)$. Furthermore, $\|\mathbb{M}\|_F^2 = \lambda_1 \sum_{i>1} \lambda_i$. Thus Eq 9 reduces to checking if*

$$\frac{\lambda_1^{3/2} (\sum_{j>1} \lambda_j)^3}{\left( \sum_i \lambda_i^2 \right)^{3/2}} = o(n)$$

**Remark 2** (Coordinates with summable sub-Gaussian parameters). *Eq 9 imposes a growth condition on the moments of both the data and a Gaussian analog. One setting for which both growth rates are in fact bounded is if the coordinates of $X$ are sub-Gaussian and the sub-Gaussian parameters satisfy $\sum_{i=1}^d \nu_i < C < \infty$ following similar arguments to Proposition 1.*

**Remark 3** (Constant vs Adaptive Learning Rate). *Adaptive learning rates are also commonly studied in the literature on Oja's algorithm and have the advantage that they require no prior knowledge of the sample size. It should be noted that our results hold for a wide range of learning rates, ranging from $\log(nd) \ll \eta_n \ll n^{1/3}$, so our results will still apply so long as in the initial guess of the sample size is not off by orders of magnitude. We leave a detailed study of the adaptive learning rate setting to future work.*

As a corollary of our main theorem, we obtain the following error bound on the $\sin^2$ error.

**Corollary 1.** *Under the conditions in Theorem 1, we have*

$$\sin^2(\hat{v}_1, v_1) = O_P\left( \frac{\eta_n M_d}{n(\lambda_1 - \lambda_2)} \right)$$

**Remark 4** (Comparison with previous work). *As a byproduct of our analysis, we recover the sharpest convergence rates for Oja's algorithm in the literature. If we set $\eta_n = c_1 \log nd/(\lambda_1 - \lambda_2)$, for large enough $c_1$, the dominating term in the error is $O_P\left( \frac{M_d \log nd}{n(\lambda_1 - \lambda_2)^2} \right)$ under mild conditions on $d$. This matches the bound in [25].*

**Remark 5** (Rate of convergence in Kolmogorov distance). *To simplify the theorem statement, we have stated Theorem 1 without giving an explicit rate of convergence in the Kolmogorov distance. Convergence rates depend on the rate of decay of the remainder terms, which are worked out in Supplement Section B.3, and the magnitude of the quantity in Eq 9. The contribution of the latter quantity to the rate is worked out in the IID case in [51].*

**Remark 6** (Lower bound on norm). *While our rate matches the sharp bounds in literature and our assumptions on norm upper bounds are similar or weaker than previous work, we do assume a lower bound on the Frobenius norm of the covariance matrix as in Eq 8. Note that if indeed all $X_i$'s were a scalar multiple of $v_1$, then the $\bar{\mathbb{V}}_n$ matrix in Eq 7 will be zero. This will lead to a perfect point estimate, but there will not be any variability from the data and hence there will be no non-degenerate approximation. The lower bound on the norm is not resulting from loose analysis. Similar lower bounds on the variance are imposed in the high-dimensional CLT literature [7, 8].*

Now we provide a proof sketch of Theorem 1 below.

*Proof sketch for Theorem 1.* We provide the main steps in our derivation. The detailed calculations can be found in Supplement Section B.

1. We start by expressing the $\sin^2$ error as a quadratic form:

$$
\begin{aligned}
\sin^2(v_1, \hat{v}_1) &= 1 - \frac{u_0^T B_n^T v_1 v_1^T B_n u_0}{u_0^T B_n^T B_n u_0} = \frac{u_0^T B_n^T (I - v_1 v_1^T) B_n u_0}{u_0^T B_n^T B_n u_0} \\
&= \frac{(V_\perp V_\perp^T B_n u_0)^T (V_\perp V_\perp^T B_n u_0)}{\|B_n u_0\|^2}
\end{aligned}
\tag{11}
$$

where in the last line we used the fact that $V_\perp V_\perp^T$ is idempotent. Our proof strategy for the central limit theorem involves further approximating Eq 11 with an inner product of the Hájek projection (first-order) term in Eq 5.

2. Our second step is to show that $\|B_n u_0\|$ concentrates around its expectation $(1 + \eta_n \lambda_1 / n)^n |v_1^T u_0|$.

3. Next we establish that $\frac{\|V_\perp V_\perp B_n V_\perp V_\perp^T u_0\|_2}{\|B_n u_0\|}$ is $O_P\left(\sqrt{d} \cdot \exp\{-\eta_n(\lambda_1 - \lambda_2)\} + \sqrt{\frac{\eta_n^3 M_d^2 \log d}{n^2}}\right)$.

   This is achieved by using a similar recursive argument as in [25], but with the crucial observation that the residual or common difference term is of a lower order because it can be replaced by a matrix product minus its expectation.

4. Now we go back to the expansion in Eq 5.

$$
(v_1^T u_0) V_\perp V_\perp^T B_n v_1 = (v_1^T u_0) \sum_k V_\perp V_\perp^T T_k v_1
$$

   Since $T_0 = (I + \eta_n/n\Sigma)^n$, $V_\perp V_\perp^T T_0 v_1$ is the zero vector. Now we examine the $(v_1^T u_0) V_\perp V_\perp^T (B_n - T_1) v_1$ term. Here we use the structure of the higher order terms $T_k$. In particular, we use the fact that it is a matrix product interlaced with $k$ $X_i X_i^T - \Sigma$ matrices. For example, for $k = 2$ we have

$$
T_2 = \frac{\eta_n^2}{n^2} \sum_{i<j} \left(I + \frac{\eta_n}{n}\Sigma\right)^{i-1} Y_i \left(I + \frac{\eta_n}{n}\Sigma\right)^{j-i-1} Y_j \left(I + \frac{\eta_n}{n}\Sigma\right)^{n-j}
$$

   We show that the norm of $(v_1^T u_0) V_\perp V_\perp^T (B_n - T_1) v_1$, normalized by the denominator, is $O(\eta_n^2 M_d^2/n^2)$. The fact that the summands of $T_k$ are uncorrelated and $T_k$ and $T_\ell$ are uncorrelated for $k \neq \ell$ makes this possible.

5. Finally, we are left with $V_\perp V_\perp^T T_1 v_1 (v_1^T u_0)$. Note that this is of the following form:

$$
\frac{\eta_n}{n} \frac{(v_1^T u_0) V_\perp V_\perp^T T_1 v_1}{|v_1^T u_0|(1 + \lambda_1 \eta_n/n)^n} = \frac{\eta_n \operatorname{sgn}(v_1^T u_0)}{n} \sum_{i=1}^n V_\perp \Lambda_\perp^{i-1} V_\perp^T (X_i X_i^T - \Sigma) v_1
$$

   It is not hard to see that this is a sum of independent random vectors with covariance matrix $\eta_n/n \bar{\mathbb{V}}_n$ (see Eq 7).

6. We adapt a result of distributional convergence of squared norm of sums of IID random vectors in [51] to squared norm of sums of independent random vectors. Under the assumptions 9 and 8, the conditions of distributional convergence are satisfied.

7. Finally, all the error terms are combined along with an anti-concentration argument for $\chi^2$ to establish the final result. The full proof and accompanying lemmas are in Section B of the Supplement.

$\square$

## 3.2 Bootstrap consistency

Using the weighted $\chi^2$ approximation for inference requires estimating the eigenvalues of $\Sigma$ and other population quantities; however, accurate estimates may not be available in a streaming setting. Instead, we propose a streaming bootstrap procedure that mimics the properties of the original Oja algorithm. While a similar structure leads to error terms that are similar to the CLT, the analysis of the bootstrap presents its own technical challenges. In what follows let $P^*$ denote the bootstrap measure, which is conditioned on the data, and let $\mathbb{E}^*[\cdot]$ denote the corresponding expectation operator.

A common strategy for establishing consistency of the Gaussian multiplier bootstrap is to invoke a Gaussian comparison lemma. Since the multipliers are themselves Gaussian and the data is treated as fixed, the idea is that one can use specialized results for comparing the distributions of two Gaussians (bootstrapped $Z_n^*$ and approximating $Z_n$ from the CLT) that only depend on how close the covariance matrices $\mathbb{E}^*[Z_n^* Z_n^{*T}]$ and $\mathbb{E}[Z_n Z_n^T]$ are in an appropriate metric. Using a Gaussian comparison lemma for quadratic forms (see Supplement Section C.3), we have the following result for the bootstrapped $\sin^2$ error:

**Lemma 1.** *[Bounding the difference between the bootstrap covariance and true covariance] Let:*

$$Z_n^* = sgn(v_1^T u_0)\sqrt{\frac{\eta_n}{n}} \sum_i W_i V_\perp \Lambda_\perp^{i-1} V_\perp^T (X_i X_i^T - X_{i-1} X_{i-1}^T) v_1. \tag{12}$$

*Recall the definition of $\bar{\mathbb{V}}_n$ from Eq 7. We have,*

$$|\text{trace}(\mathbb{E}^*[Z_n^* Z_n^{*T}] - \bar{\mathbb{V}}_n)|, \|\mathbb{E}^*[Z_n^* Z_n^{*T}] - \bar{\mathbb{V}}_n\|_F = O_P\left(\sqrt{\frac{\mathbb{E}\|X_1 X_1^T - \Sigma\|^4}{n(\lambda_1 - \lambda_2)}}\right)$$

With this lemma in hand, we are ready to state our bootstrap result.

**Theorem 2** (Bootstrap Consistency). *Suppose that the conditions of Theorem 1 are satisfied. Furthermore, let $\alpha_n$ be a sequence such that $P(\mathcal{A}_n^c) \to 0$, where $\mathcal{A}_n$ is defined as $\mathcal{A}_n = \left\{\max_{i \leq i \leq n} \|X_i\|^2 \leq \alpha_n\right\}$. Further suppose that $\frac{M_d \log^2 d \, \eta_n^2}{n} \to 0$, $\frac{(\alpha_n^3 \vee M_d \log d) \, \alpha_n \eta_n^3}{n} \to 0$, $\frac{\alpha_n M_d \eta_n^2}{n(\lambda_1 - \lambda_2)} \to 0$, and $\frac{\mathbb{E}[\|X_1 X_1^T - \Sigma\|^4]}{n(\lambda_1 - \lambda_2)} \to 0$. Then,*

$$\sup_{t \in \mathbb{R}} \left| P^*(n/\eta_n \sin^2(v_1^*, \hat{v}_1) \leq t) - P(n/\eta_n \sin^2(\hat{v}_1, v_1) \leq t) \right| \xrightarrow{P} 0$$

*Proof sketch of Theorem 2.* The proof follows a similar route to Theorem 2. We provide a detailed analysis in Supplementary Section. We use a bootstrap version of the Hoeffding decomposition conditioned on the data, stated in Supplement Section. In step one we have $B_n^*$ replace $B_n$, where $B_n^*$ is given by:

$$B_n^* = \prod_{i=1}^{n} \left(I + \eta_n/n(X_i X_i^T + W_i(X_i X_i^T - X_{i-1} X_{i-1}^T))\right)$$

We work out Step 1 using concentration of matrix products [23]. For steps 2-3, we see that $T_k^*$ has the same structure as $T_k$ with the difference that $(I + \eta_n \Sigma/n)^i$ is replaced by its sample counterpart which is a product of $i$ independent matrices of the form $I + \eta_n/n X_j X_j^T$. Concentration of these terms in operator norm are established with results from [23]. Finally for step 4, we see that the main term that approximates the bootstrap residual $\widehat{V}_\perp \widehat{V}_\perp^T B_n^* u_0$ is given by $\sqrt{\eta_n/n} Z_n^*$, where $Z_n^*$ is given in Eq 12. Conditioned on the data, this is already Normally distributed since the multiplier random variables $W_i$ are themselves Gaussian. We then invoke the Gaussian comparison result Lemma 1 to obtain convergence to the weighted $\chi^2$ approximation. $\square$

We now make a couple of points regarding our analysis. It should be noted that the terms in the product are weakly dependent, which is different from the CLT and would seem to complicate concentration arguments used to establish bootstrap consistency. However, the dependence is not strong and second-moment methods may be used. We also operate on a good set in which the norms

of the the updates are not too large, which is far less restrictive than assuming an almost sure bound on the norm.

In theorem above, we have stated the good set $\mathcal{A}_n$ in an abstract manner, but one may wonder how stringent the condition is in various problem settings. Below, we describe a general setup with sub-Gaussian entries of $X_i$ in which $\alpha_n$ grows as $\log n$; under milder forms of various decay, all we need is for $\alpha_n$ to grow slowly with $n$. Here $\|\cdot\|_{\psi_1}$ is the sub-Exponential Orlicz norm and $\|\cdot\|_{\psi_2}$ is the sub-Gaussian Orlicz norm (see, for example [50]).

**Proposition 1** (The effect of variance decay on the norm). *For each $1 \leq j \leq p$, suppose that $X_{1j}$ satisfies $\|X_{1j}\|_{\psi_2} \leq \nu_j \sum_{j=1}^{p} \nu_j \leq C_1 < \infty$. Then, for some universal constant $C_2 > 0$, $\left\|\sum_{j=1}^{p}(X_{1j}^2 - \mathbb{E}[X_{1j}^2])\right\|_{\psi_1} < C_2$, and for some $c_1, c_2 > 0$,*

$$P\left(\max_{1 \leq i \leq n}\|X_i\|^2 > c_1 \log n\right) \leq \frac{c_2}{n}$$

We now present experimental validation of our bootstrap procedure below.

## 4   Experimental validation of the online multiplier bootstrap

We draw $Z_{ij} \overset{IID}{\sim} \text{Uniform}(-\sqrt{3}, \sqrt{3})$, for $i = 1, \ldots, n$ and $j = 1, \ldots d$. Consider a PSD matrix $K_{ij} = \exp(-|i - j|c)$ with $c = 0.01$. We create a covariance matrix such that $\Sigma_{ij} = K(i, j)\sigma_i\sigma_j$. We consider $\sigma_i = 5i^{-\beta}$ for $\beta = 0.2$ and $\beta = 1$. Now we transform the data to introduce dependence by letting $X_i = \Sigma^{1/2}Z_i$. By construction, we have that $\mathbb{E}[X_iX_i^T] = \Sigma$ for all $1 \leq i \leq n$. Our goal is to simply demonstrate that the bootstrap distribution of $\sin^2$ errors closely match that of the sampling distribution. To this effect, we fix $u_0$ and draw 500 datasets and run streaming PCA on each and then construct an empirical CDF ($F$) from the $\sin^2$ error with the true $v_1$. This is the point of comparison for the bootstrap distribution ($F^*$), for which we fix a dataset $X$. We then invoke algorithm 1 to obtain 500 bootstrap replicates $\hat{v}_1^*$ as well as the Oja vector for the dataset $\hat{v}_1$. The bootstrap distribution is the empirical CDF of $1 - (\hat{v}_1^T\hat{v}_1^*)^2$. We use $\eta_n = \log n$. In Figure 1, we see that for $\beta = 0.2$ (see (A) and (B)), where the variance decay is slow and therefore the error bounds of the residual terms are expected to be large, the quality of approximation is poorer compared to (C) and (D), where $\beta = 1$. However, even for $\beta = 0.2$, increasing $n$ improves performance. Also note that, for (A) and (B) the variance decay does not satisfy our theorems conditions and thus, the normalized error does not behave like a $O_P(1)$ random variable. However, for (C) and (D) the variance decay satisfies the conditions and in this case the normalized error is $O_P(1)$, which happens to be in the [0,1] range for this example.

## 5   Discussion

Modern tools in non-asymptotic random matrix theory have given rise to recent breakthroughs in establishing pointwise convergence rates for stochastic iterative methods in optimizing certain nonconvex objectives, including the classic Oja's algorithm for online principal component analysis. By synthesizing modern random matrix theory tools with classic results from the U-statistics literature and recently developed high-dimensional central limit theorems, we extend the error analysis of Oja's algorithm from pointwise convergence rates to *distributional* convergence and moreover establish an efficient online bootstrap method for Oja's algorithm to quantify the error on the fly. Our results are a first step toward incorporating uncertainty estimation into the general framework of stochastic optimization algorithms, but we acknowledge the present limitations of our analysis: new tools will be needed to extend the current analysis to estimating higher-dimensional principal subspaces, and additional tools will be needed to account for non-independent matrix products which appear beyond the setting of online PCA.

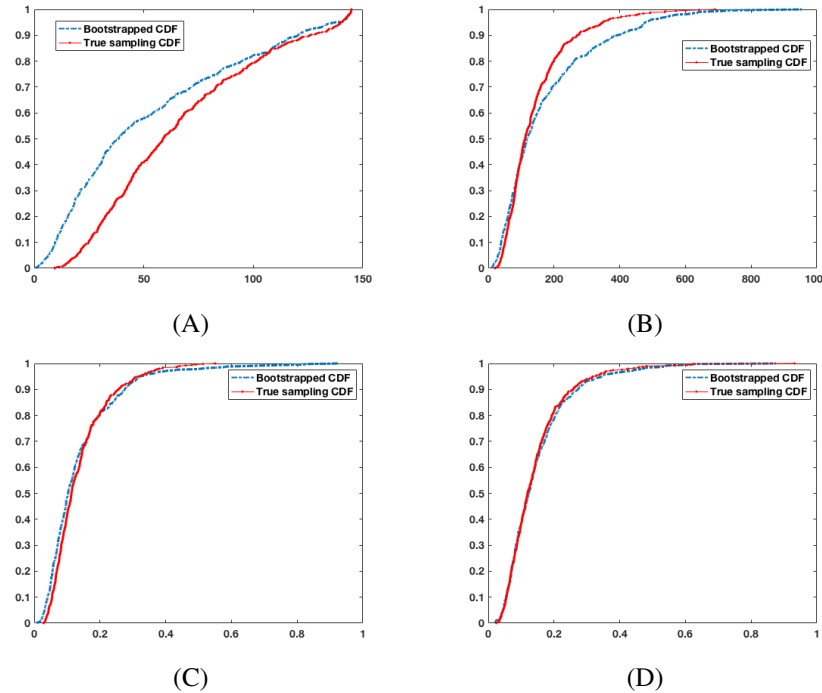

Figure 1: Bootstrapped and sampling CDF for $n = 1000, d = 500$ in (A) and (C) and for $n = 10,000, d = 500$ in (B) and (D). (A) and (B) use $\beta = 0.2$ whereas (C) and (D) use $\beta = 1$.

## Acknowledgment

P.S. and R.L. are supported in part by NSF 2019844 and NSF HDR-1934932. R.W. is supported in part by AFOSR MURI FA9550-19-1-0005, NSF DMS 1952735, NSF HDR-1934932, and NSF 2019844.

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
