# Supplementary Material for "Bootstrapping the error of Oja's algorithm"

**Robert Lunde**
University of Michigan
rlunde@umich.edu

**Purnamrita Sarkar**
University of Texas at Austin
purna.sarkar@austin.utexas.edu

**Rachel Ward**
University of Texas at Austin
rward@math.utexas.edu

## Supplementary Material

In this document we provide the detailed proofs of results presented in the main manuscript. In Section A, we provide a proof for the Hoeffding expansion of the matrix product in Eq 5 of the main document. We also provide the Hoeffding decomposition for the bootstrap in Proposition A.4. In Section B we provide all results needed for a complete proof of Theorem 1. In Sections B.1, B.2, and B.3 we provide the proof of Theorem 1, the adaptation of high dimensional CLT of [8] to our setting and all supporting lemmas, respectively.

In Section C we provide all details of the proof of the Bootstrap consistency, i.e. Theorem 2. To be specific, Section C.1 has the proof of Theorem 2; Section C.2 has the proof of Lemma 1, Section C.3 has the statement and proof of the Gaussian comparison lemma, and Section C.4 has all the supporting lemmas. Finally, in Section D, we provide a proof of Proposition 1.

## A  On the Hoeffding decomposition

We discuss Hoeffding decompositions for a function $f$ of $n$ independent random variables $X_1, \ldots X_n$, where the random variables take values in an arbitrary space and the function takes values[1] in $\mathbb{R}^{d \times d}$ or $\mathbb{R}^d$. The following exposition largely follows [6].

With Hoeffding decompositions, we project $T(X_1, \ldots, X_n)$ onto spaces of increasing complexity that are orthogonal to each other. In our setup, orthogonality means $\langle f, g \rangle_{L^2} = 0$ where $\langle f, g \rangle_{L^2} = \int \langle f, g \rangle dP$. Here, $\langle f, g \rangle = \mathrm{Trace}(f^T g)$ in the matrix case and $\langle f, g \rangle = f^T g$ in the vector case. The first-order projection, also known as a Hájek projection, involves projecting our function onto a space of functions of the form

$$g^{(i)}(X_i)$$

---

[1]The math generalizes to Hilbert spaces due to the Hilbert projection theorem but we specialize to these cases for concreteness.

where $g^{(i)}$ satisfies $E[g^{(i)}] = 0$. We will let $H^{(i)}(X_i)$ denote the corresponding projection. Since the functions $g^{(i)}, g^{(j)}$ are mutually orthogonal for $i \neq j$, the sum of the projections is equivalent to the projection onto the space spanned by functions of the form:

$$\sum_{i=1}^{n} g^{(i)}(X_i)$$

The higher-order spaces have the form:

$$g^{(S)}(X_i : i \in S)$$

where $S \subseteq \{1, \ldots, n\}$ and the functions satisfy $\mathbb{E}[g^{(S)} \mid X_i : i \in R] = 0$ for any $R \subset S$, including $R = \emptyset$, which implies $\mathbb{E}[g^{(S)}] = 0$. If $R \not\subset S$ and $S \not\subset R$, $\langle g^{(S)}, g^{(R)} \rangle_{L^2} = 0$ since, by conditional independence given $\{X_i : i \in R \cap S\}$:

$$\mathbb{E}[\mathbb{E}[\langle g^{(S)}, g^{(R)} \rangle \mid X_i : i \in R \cap S]] = \mathbb{E}\left[ \langle \mathbb{E}[g^{(S)} \mid X_i : i \in R \cap S], \mathbb{E}[g^{(R)} \mid X_i : i \in R \cap S] \rangle \right] = 0 \tag{S.1}$$

Combining these projections leads to the following representation, known as the Hoeffding decomposition:

$$T(X_1, \ldots, X_n) = \sum_{k=0}^{n} \sum_{|S|=k} H^{(S)}(X_i : i \in S)$$

While the following proposition is stated for real-valued functions in [6][Lemma 11.11], it turns out that the proof there generalizes to our setting without difficulty due to machinery for projections in Hilbert spaces.

**Proposition A.1** (Hoeffding projections)**.** *Let* $X_1, \ldots, X_n$ *be arbitrary random variables and let suppose* $\langle T, T \rangle_{L_2} < \infty$. *Then the projection on the the space of functions of the form* $g^{(S)}(X_i : i \in S)$ *with* $\mathbb{E}[g^{(S)} \mid X_i : i \in R] = 0$ *for any* $R \subset S$ *has the form:*

$$H^{(S)}(T) = \sum_{R \subseteq S} (-1)^{|S|-|R|} \, \mathbb{E}\left[T \mid X_i : i \in R\right]$$

For completeness, we provide a proof of the proposition below.

*Proof.* We begin by verifying that the space of all random matrices (vectors) satisfying $\|A\|_{L^2} < \infty$ forms a Hilbert Space. First, it is clear that $\langle \cdot, \cdot \rangle_{L^2}$ is indeed an inner product. Linearity follows from linearity of the inner product $\langle \cdot, \cdot \rangle$ and linearity of expectations and conjugate symmetry follows from this property holding pointwise in $\Omega$ for $\langle \cdot, \cdot \rangle$. Positive definiteness again follows from the fact that this property holds pointwise in $\Omega$; then a standard contradiction argument yields that if $\langle x, x \rangle_{L^2} = 0$, but $x$ is not equal to 0 almost surely, there exists some $M$ such that for some $\delta > 0$, $P(\|x\| > \frac{1}{M}) \geq \delta$ and hence $\int \langle x, x \rangle dP \geq \delta/M > 0$, a contradiction.

One can again adapt standard arguments for completeness of $L_2$ spaces to our setting; namely, show that Cauchy sequences converging in $L_2$ implies convergence almost everywhere, and then invoke completeness of the Hilbert space over matrices/vectors along with integral convergence theorems; see for example, the proof of Theorem 1.2, page 159 in [5].

Now to verify that this function is indeed the projection, we invoke the Hilbert Projection Theorem; see for example, Lemma 4.1 of [5]. To use this theorem, we need to check that the space spanned by functions of the form $g^{(S)}$ satisfying the condition $\mathbb{E}[g^{(S)} \mid X_i : i \in R] = 0$ for any $R \subset S$ is a closed subspace. Linearity of the space follows from the fact that the sum of such functions satisfies the constraint; therefore it is a subspace. To check closure, let $\|f\|^2 = \langle f, f \rangle$ and consider some (convergent) sequence in this subspace $(g_\alpha^{(S)})_{\alpha \geq 1}$ where $g_\alpha^{(S)} \to g^{(S)}$ and observe that, for any $R \subset S$:

$$\mathbb{E}[\|g_\alpha^{(S)} - g^{(S)}\|^2] = \mathbb{E}[\ \mathbb{E}[\|g_\alpha^{(S)} - g^{(S)}\|^2 \mid X_i : i \in R]\ ]$$
$$\geq \mathbb{E}\left[\|\mathbb{E}[g_\alpha^{(S)} - g^{(S)} \mid X_i : i \in R]\|^2\right]$$
$$\geq \mathbb{E}\left[\|\mathbb{E}[g^{(S)} \mid X_i : i \in R]\|^2\right]$$

where above we used the fact that $\mathbb{E}[g_\alpha^{(S)} \mid X_i : i \in R] = 0$ for all $\alpha$ by assumption. Since the LHS converges to 0, it follows that $\mathbb{E}[g^{(S)} \mid X_i : i \in R]$ must be equal to 0 almost surely. Since the limit satisfies $\mathbb{E}[g^{(S)} \mid X_i : i \in R] = 0$ for all $R \subset S$, it belongs in the space, proving closure.

Now, we show that the stated expression is indeed the Hoeffding projection. First, to show that belongs in this space, we have, following analogous reasoning to [6], for any $C \subset A$,

$$\mathbb{E}[H^{(A)}(T) \mid X_i : i \in C] = \sum_{B \subseteq A} (-1)^{|A|-|B|} \mathbb{E}[T \mid X_i : i \in B \cap C]$$
$$= \sum_{D \subseteq C} \sum_{j=0}^{|A|-|C|} (-1)^{|A|-(|D|+j)} \binom{|A| - |C|}{j} \mathbb{E}[T \mid X_i : i \in D]$$
$$= \sum_{D \subseteq C} (-1)^{|C|-|D|} \mathbb{E}[T \mid X_i : i \in D] (1-1)^{|A|-|C|} = 0$$

where the last line follows from the Binomial Theorem. Now as a consequence of the Hilbert Projection Theorem, it suffices to show that $H^{(A)}(T)$ satisfies the property:

$$\langle T - H^{(A)}(T), g^{(A)} \rangle_{L^2} = 0$$

for any $g^{(A)}$ in the space. In the matrix case, we have

$$\langle T - H^{(A)}(T), g^{(A)} \rangle_{L^2} = \sum_{j=1}^{d} \sum_{k=1}^{d} \mathbb{E}[(T_{jk} - \mathbb{E}[T_{jk} \mid X_i : i \in A]) \cdot g_{jk}^{(A)}]$$
$$+ \sum_{j=1}^{d} \sum_{k=1}^{d} \sum_{B \subset A} \mathbb{E}\left[(-1)^{|A|-|B|} \mathbb{E}[T_{jk} \mid X_i : i \in B] \cdot \mathbb{E}[g_{jk}^{(A)} \mid X_i : i \in B]\right]$$

The first term above is 0 since conditional expectations may be viewed as an orthogonal projection in the Hilbert Space with inner product $\int fg\, dP$ into the closed subspace of $\sigma(X_i : i \in A)$-measurable functions. The second term is zero since $\mathbb{E}[g_{jk}^{(A)} \mid X_i : i \in B] = 0$ for any $B \subset A$. The vector case is analogous.

Since this property holds, it must be the unique (up to measure 0 sets) minimizer and projection. □

Now an immediate corollary for our setting follows.

**Proposition A.2** (Orthogonality of Hoeffding projections). *Let:*

$$B_n = \sum_{k=0}^{n} \sum_{|S|=k} H^{(S)}$$

*where $A^{(S)}$ is the Hoeffding projection corresponding to the set $S \subseteq \{1, \ldots, n\}$. Then,*

$$\mathbb{E}\left[\|B_n\|_F^2\right] = \sum_{k=0}^{n} \sum_{|S|=k} \mathbb{E}\left[\|A^{(S)}\|_F^2\right]$$

$$\mathbb{E}\left[\|B_n x\|^2\right] = \sum_{k=0}^{n} \sum_{|S|=k} \mathbb{E}\left[\|A^{(S)} x\|^2\right]$$

*where the last inequality holds for all $x \in \mathbb{R}^d$.*

*Proof.* Letting $g^{(S)} = H^{(S)}$ and $g^{(R)} = H^{(R)}$ in Eq S.1, we have that $\langle H^{(S)}, H^{(R)} \rangle_{L^2} = 0$ for all $R \neq S$ and the result follows. $\square$

It remains to be shown that Hoeffding decomposition has the form stated in Eq 5. Deriving all projections in the Hoeffding decomposition for a general function is typically non-trivial, but the product structure facilitates our proof below. Before establishing the Hoeffding decomposition, following for example, [1] observe that the following inverse relation holds:

**Proposition A.3** (Conditional expectation and Hoeffding projections).

$$\mathbb{E}\left[T \mid X_i : i \in S\right] = \sum_{R \subseteq S} H^{(R)}(T)$$

*Proof.* Observe that:

$$\mathbb{E}[T \mid X_i : i \in S] = \sum_{k=0}^{n} \sum_{|R|=k} \mathbb{E}[H^{(R)}(T) \mid X_i : i \in S]$$

Since the conditional expectation is zero for $R \not\subseteq S$ and for $R \subseteq S$, the Hoeffding projection is fixed, the result follows. $\square$

Now we are ready to establish the form of the Hoeffding projection for any $S \subseteq \{1, \ldots, n\}$. We in fact prove a slightly stronger statement, which makes the induction argument more natural. In what follows let $S[i]$ denote the $i$th element in $S$. We will also use $H^{(S)}$ instead of $H^{(S)}(T)$ when it is clear from the context.

**Theorem A.1** (Hoeffding projections for Oja's algorithm). *Define:*

$$T_{-j} = \prod_{i=j+1}^{n} \left(I + \frac{\eta_n}{n} X_i X_i^T\right), \quad T = T_{-0} = \prod_{i=1}^{n} \left(I + \frac{\eta_n}{n} X_i X_i^T\right),$$

*Then for any $S \subseteq \{1, \ldots, n\}$ and for all $0 \leq j < S[1]$, we have the Hoeffding projection of $T_{-j}$ onto $\{X_i : i \in S\}$ may be expressed as:*

$$H_{-j}^{(S)} = \prod_{i=j+1}^{n} A_i^{(S)}, \quad H^{(S)} = H_{-0}^{(S)} \tag{S.2}$$

*where:*

$$A_i^{(S)} = \begin{cases} \frac{\eta_n}{n}(X_i X_i^T - \Sigma) & i \in S \\ I + \frac{\eta_n}{n}\Sigma & i \notin S \end{cases}$$

*Proof.* We will conduct (strong) induction on $k = |R|$, where $R \subseteq S$. We will start with the base case $k = 1$; $k = 0$ is simply the expectation. For the base case $|R| = 1$, a direct calculation is possible, since:

$$H_{-j}^{(R)} = \mathbb{E}[T_{-j} \mid X_i : i \in R] - \mathbb{E}[T_{-j}],$$

which has the stated form. Now, we will suppose that the inductive hypothesis holds. In what follows, let $S[1] = k$ and define the conditional expectation for any set $S$ as:

$$\mathbb{E}\left[T_{-j} \mid X_i : i \in S\right] = \prod_{i=j+1}^{n} E_i^{(S)},$$

where:

$$E_i^{(S)} = \begin{cases} I + \frac{\eta_n}{n}X_i X_i^T & i \in S \\ I + \frac{\eta_n}{n}\Sigma & i \notin S \end{cases}$$

We will now add and subtract a product where an entry corresponding to $S[1]$ in $\mathbb{E}[T_{-j} \mid X_i : i \in S]$ is replaced by $(I + \frac{\eta_n}{n}\Sigma)$. Doing, so we have

$$\mathbb{E}[T_{-j} \mid X_i : i \in S] = \mathbb{E}\left[T_{-j} \mid X_i : i \in S\right] - (I + \frac{\eta_n}{n}\Sigma)^{k-j} \times \prod_{i=k+1}^{n} E_i^{(S)}$$

$$+ (I + \frac{\eta_n}{n}\Sigma)^{k-j} \times \prod_{i=k+1}^{n} E_i^{(S)}$$

We recognize the second summand as $\mathbb{E}[T_{-j} \mid X_i : i \in S_{-k}]$, where $S_{-k} = \{i \in S, i \neq k\}$. Now for the first summand, taking the difference we have the term

$$(I + \frac{\eta_n}{n}\Sigma)^{k-j-1} \times \frac{\eta_n}{n}(X_k X_k^T - \Sigma) \times \prod_{i=k+1}^{n} E_i^{(S)}$$

$$= (I + \frac{\eta_n}{n}\Sigma)^{k-j-1} \times \frac{\eta_n}{n}(X_k X_k^T - \Sigma) \times \mathbb{E}\left[T_{-k} \mid X_i : i \in S_{-k}\right]$$

By Proposition A.3, we may represent a conditional expectation as:

$$\mathbb{E}\left[T_{-k} \mid X_i : i \in S_{-k}\right] = \sum_{R \subseteq S_{-k}} H_{-k}^{(R)} \tag{S.3}$$

Furthermore, by the inductive hypothesis, each $H_{-k}^{(R)}$ takes the form in Eq S.2. Now, combining the two parts, we have

$$\mathbb{E}[T_{-j} \mid X_i : i \in S] = \sum_{R \subseteq S_{-k}} (I + \frac{\eta_n}{n}\Sigma)^{k-j-1} \times \frac{\eta_n}{n}(X_k X_k^T - \Sigma) \times H_{-k}^{(R)}$$
$$+ \sum_{R \subseteq S_{-k}} (I + \frac{\eta_n}{n}\Sigma)^{k-j} \times H_{-k}^{(R)}$$
$$= \prod_{i=j+1}^{n} A_i^{(S)} + \sum_{R \subset S} H_{-j}^{(R)}$$

For the last step, notice that with the exception of $R = S_{-k}$ in the first sum, each product in the sum corresponds to a Hoeffding projection of some set of size less than $k$ by the inductive hypothesis. The first term must be the Hoeffding projection onto $S$ (with $S[1] = k > j$) by the same argument as Eq S.3, i.e.

$$H_{-j}^{(S)} = \prod_{i=j+1}^{n} A_i^{(S)},$$

proving the desired result. □

Now, since the Hoeffding decomposition is a sum of Hoeffding projections by definition, we have the following corollary.

**Corollary A.1** (Hoeffding decomposition for Oja's algorithm).

$$B_n = \sum_{k=0}^{n} \sum_{|S|=k} H^{(S)}$$

where $A^{(S)}$ is given by $H^{(S)}$ in Eq S.2.

It turns out that the bootstrap Hoeffding decomposition can be proved using the same strategy in Theorem A.1, where $X_1, \ldots, X_n$ is treated as fixed in the bootstrap measure. We state the result below.

**Proposition A.4** (Hoeffding decomposition for the bootstrap).

$$B_n^* = \sum_{k=0}^{n} \sum_{|S|=k} \alpha^{(S)}$$

where $\alpha^{(S)} = \prod_{i=1}^{n} \alpha_i^{(S)}$ and $\alpha_i^{(S)}$ is given by:

$$\alpha_i^{(S)} = \begin{cases} \frac{\eta_n}{n} W_i \cdot (X_i X_i^T - X_{i-1} X_{i-1}^T) & \text{if } i \in S \\ I + \frac{\eta_n}{n} X_i X_i^T & \text{otherwise} \end{cases}$$

# B  Central limit theorem for Oja's algorithm

## B.1  Proof of Theorem 1

*Proof of Theorem 1.* Our strategy will be to approximate $\sin^2$ distance for estimated eigenvector with a quadratic form, and invoke a high-dimensional central limit theorem result. The remainder terms will be bounded using an anti-concentration result for weighted $\chi^2$ random variables due to [8].

Observe that $\sin^2(\hat{v}_1, v_1)$ has the representation:

$$1 - \left( v_1^T \frac{B_n u_0}{\|B_n u_0\|} \right)^2 = \frac{u_0^T B_n^T (I - v_1 v_1^T) B_n u_0}{\|B_n u_0\|^2}$$

Let $V_\perp V_\perp^T = I - v_1 v_1^T$. Clearly, $V_\perp V_\perp^T$ is idempotent and is a projection matrix, implying that it is also symmetric. Therefore,

$$\frac{n}{\eta_n} \cdot \sin^2(u_n, v_1) = \frac{(\sqrt{n/\eta_n} V_\perp V_\perp^T B_n u_0)^T (\sqrt{n/\eta_n} V_\perp V_\perp^T B_n u_0)}{\|B_n u_0\|^2} \tag{S.4}$$

Let $a_1 = (v_1^T u_0)$ denote the scalar projection of $u_0$ so that $u_0 = a_1 v_1 + w$, where $w$ is in the orthogonal complement of $v_1$.

Our first reduction of (S.4) is to approximate the denominator with a more convenient quantity. By Lemma B.2, we have that (S.4) may be written as

$$\frac{(\sqrt{n}/\eta_n \cdot V_\perp V_\perp^T B_n u_0)^T (\sqrt{n}/\eta_n \cdot V_\perp V_\perp^T B_n u_0)}{a_1^2 (1 + \frac{\eta_n}{n} \lambda_1)^{2n}} \cdot R_1$$

where

$$R_1 = \frac{\|B_n u_0\|^2}{a_1^2 (1 + \frac{\eta_n}{n} \lambda_1)^{2n}} = 1 - O_P \left( \sqrt{d} \exp\left( -\frac{\eta_n}{2} (\lambda_1 - \lambda_2) \right) + \sqrt{\frac{\eta_n^2 M_d \log d}{n}} \right)$$

While the aforementioned Lemma is stated for $\frac{\|B_n u_0\|}{|a_1|(1 + \frac{\eta_n}{n} \lambda_1)^n}$, the relationship holds for the squared quantity since with high probability for $n$ large enough, $|\frac{\|B_n u_0\|}{|a_1|(1 + \frac{\eta_n}{n} \lambda_1)^n}| \leq 2$ and $|x^2 - 1^2| \leq 3|x - 1|$ for all $-2 \leq x \leq 2$.

We will further approximate the quantity $\sqrt{n}/\eta_n \cdot V_\perp V_\perp^T B_n u_0$. First we will bound the contribution of $V_\perp V_\perp^T B_n V_\perp V_\perp^T$. By Lemma B.3 we have that:

$$R_2 := \sqrt{\frac{n}{\eta_n}} \cdot \frac{V_\perp V_\perp^T B_n V_\perp V_\perp^T u_0}{|a_1|(1 + \frac{\eta_n \lambda_1}{n})^n} = O_P \left( \sqrt{\frac{nd}{\eta_n}} \cdot \exp\{-\eta_n (\lambda_1 - \lambda_2)\} + \sqrt{\frac{\eta_n^2 M_d^2 \log d}{n}} \right)$$

Now it remains to bound the term $V_\perp V_\perp^T B_n v_1 (v_1^T u_0)$. First, by Corollary A.1, $B_n$ can be decomposed as:

$$B_n = \sum_{k=0}^{n} T_k$$

where for $S \subseteq \{1, \ldots, n\}$, $T_k$ is defined as:

$$T_k = \sum_{|S|=k} A^{(S)} \tag{S.5}$$

with $A^{(S)}$ taking the form in Eq S.2.

Since $v_1$ is orthogonal to $V_\perp$:

$$\sqrt{\frac{n}{\eta_n}} \cdot \frac{V_\perp V_\perp^T T_0 \; v_1 a_1}{|a_1|(1 + \eta_n/n\lambda_1)^n} = \sqrt{\frac{n}{\eta_n}} \cdot \text{sign}(a_1)(I - v_1 v_1^T)v_1 = 0.$$

Furthermore, by Lemma B.4, since $\frac{\eta_n^3 M_d^2}{n} \to 0$ by assumption,

$$R_3 := \sqrt{\frac{n}{\eta_n}} \cdot \frac{V_\perp V_\perp^T (B_n - T_1)v_1 a_1}{|a_1|(1 + \eta_n/n\lambda_1)^n} = O_P\left(\sqrt{\frac{\eta_n^3 M_d^2}{n}}\right) \tag{S.6}$$

Now our term of interest is given by:

$$\frac{(\sqrt{n/\eta_n} \cdot V_\perp V_\perp^T T_1 v_1)^T (\sqrt{n/\eta_n} \cdot V_\perp V_\perp^T T_1 v_1)}{(1 + \frac{\eta_n}{n}\lambda_1)^{2n}} \tag{S.7}$$

Now, observe that $(I + \frac{\eta_n}{n}\Sigma)$ and $v_1 v_1^T$ share a common eigenspace and therefore commute. Therefore, the terms in the product to the left of $T_1$ may be written as:

$$\frac{V_\perp V_\perp^T (I + \frac{\eta_n}{n}\Sigma)^{i-1}}{(1 + \frac{\eta_n}{n}\lambda_1)^{i-1}} = \sum_{j=2}^{d} \left(\frac{1 + \frac{\eta_n}{n}\lambda_j}{1 + \frac{\eta_n}{n}\lambda_1}\right)^{i-1} v_j v_j^T := D_{i-1}, \quad \text{say.} \tag{S.8}$$

Hence,

$$\sqrt{\frac{n}{\eta_n}} \cdot \frac{V_\perp V_\perp^T T_1 v_1}{(1 + \frac{\eta_n}{n}\lambda_1)^n} = \sqrt{\frac{\eta_n}{n}} \sum_{i=1}^{n} \left(1 + \frac{\eta_n}{n}\lambda_1\right)^{-1} D_{i-1}(X_i X_i^T - \Sigma)v_1$$

$$= S_n = \sqrt{n}\left(1 + \frac{\eta_n}{n}\lambda_1\right)^{-1} \frac{1}{n}\sum_{i=1}^{n} U_i, \quad \text{say,}$$

where

$$U_i = D_{i-1}(X_i X_i^T - \Sigma)v_1. \tag{S.9}$$

Observe that $S_n$ is a sum of independent but non-identically distributed random variables with mean 0. Therefore, if the conditions of Proposition B.1 are satisfied, we may approximate $S_n^T S_n$ with $Z_n^T Z_n$, where $\mathbb{E}[Z_n] = 0$, $\text{Var}(Z_n) = \text{Var}(S_n)$. Below define $\tilde{Z}_i$ to be a Gaussian vector with $\text{Var}(\tilde{Z}_i) = \text{Var}((X_i X_i^T - \Sigma)v_1)$. Now define $Z_i = D_{i-1}\tilde{Z}_i$. We now verify these conditions.

First, we derive a lower bound on $\left\|\bar{\mathbb{V}}_n\right\|_F$ that will be used in all of the following bounds. Observe that $\left\|\bar{\mathbb{V}}_n\right\|_F = \frac{\eta_n}{n}\left\|\sum_i \Lambda_\perp^{i-1}\mathbb{M}\Lambda_\perp^{i-1}\right\|_F$ and the $kl$th entry of $\sum_i \Lambda_\perp^{i-1}\mathbb{M}\Lambda_\perp^{i-1}$ is lower bounded by:

$$
\begin{aligned}
&\frac{\eta_n}{n}\sum_{i\geq 1}\left(\frac{1+\eta_n\lambda_{k+1}/n}{1+\eta_n\lambda_1/n}\right)^{i-1}\left(\frac{1+\eta_n\lambda_{\ell+1}/n}{1+\eta_n\lambda_1/n}\right)^{i-1}\mathbb{M}(k,\ell) \\
&\geq \frac{1-\exp(-2\eta_n(\lambda_1-\lambda_2))\left(1-\frac{\eta_n^2\lambda_1^2}{n}\right)^{-2}}{2\lambda_1-(\lambda_{k+1}+\lambda_{k+1})+\frac{\eta_n}{n}(\lambda_1^2-\lambda_k\lambda_l)}\mathbb{M}(k,\ell) \\
&\geq \frac{1-\exp(-2\eta_n(\lambda_1-\lambda_2))\left(1-\frac{\eta_n^2\lambda_1^2}{n}\right)^{-2}}{2\lambda_1+\frac{\eta_n}{n}\lambda_1^2}\mathbb{M}(k,\ell) \\
&\geq \frac{c}{\lambda_1}\ \mathbb{M}(k,\ell)
\end{aligned} \tag{S.10}
$$

for some $c > 0$ and $n$ large enough since $\exp(-\eta_n(\lambda_1-\lambda_2)) \to 0$.

For the first term of $L_q$, $q = 3$ we have

$$
\begin{aligned}
L_{3,1}^U &\leq \frac{1}{\sqrt{n}}\max_i \frac{\mathbb{E}(U_i^T\bar{\mathbb{V}}_nU_i)^{3/2}}{\|\bar{\mathbb{V}}_n\|_F^3} \\
&\leq \frac{M_d^{3/2}}{\sqrt{n}}\frac{\mathbb{E}\|V_\perp^T(X_iX_i^T-\Sigma)v_1\|^3}{\|\bar{\mathbb{V}}_n\|_F^3} \qquad \text{Since } \|\bar{\mathbb{V}}_n\| \leq M_d\eta_n \text{ from Eq 7} \\
&\leq C\frac{M_d^{3/2}\eta_n^3\lambda_1^3}{\sqrt{n}}\mathbb{E}\left(\frac{\|V_\perp^TX_1X_1^Tv_1\|}{\|\mathbb{M}\|_F}\right)^3
\end{aligned}
$$

Similarly, for the Gaussian analog, we have that:

$$
\begin{aligned}
L_{3,1}^Z &\leq \frac{1}{\sqrt{n}}\max_i \frac{\mathbb{E}(Z_i^T\bar{\mathbb{V}}_nZ_i)^{3/2}}{\|\bar{\mathbb{V}}_n\|_F^3} \\
&\leq \frac{M_d^{3/2}\eta_n^{3/2}}{\sqrt{n}}\max_i \frac{\mathbb{E}\|Z_i\|^3}{\|\bar{\mathbb{V}}_n\|_F^3} \\
&\leq \frac{M_d^{3/2}\eta_n^{3/2}}{\sqrt{n}}\frac{\mathbb{E}\|\tilde{Z}_i\|^3}{\|\bar{\mathbb{V}}_n\|_F^3} \\
&\leq C\frac{M_d^{3/2}\eta_n^3\lambda_1^3}{\sqrt{n}}\mathbb{E}\left(\frac{\|\tilde{Z}_1\|}{\|\mathbb{M}\|_F}\right)^3
\end{aligned}
$$

For the second term, using the definition of $U_i$ in Eq S.9 we have:

$$
\begin{aligned}
L_{3,2}^U &\leq \frac{1}{n}\max_{i<j}\frac{E|U_i^TU_j|^3}{\|\bar{\mathbb{V}}_n\|_F^3} \\
&= \frac{1}{n}\max_{i<j}\frac{E|v_1^T(X_iX_i^T-\Sigma)D_{i+j-2}(X_jX_j^T-\Sigma)v_1|^3}{\|\bar{\mathbb{V}}_n\|_F^3}
\end{aligned}
$$

$$\leq \frac{1}{n} \frac{\left(\mathbb{E}\|V_\perp^T(X_iX_i^T-\Sigma)v_1\|^3\right)^2}{\|\bar{\mathbb{V}}_n\|_F^3} \leq \frac{\eta_n^3\lambda_1^3}{n} \frac{\left(\mathbb{E}\|V_\perp^T(X_iX_i^T)v_1\|^3\right)^2}{\|\mathbb{M}\|_F^3}$$

For $K_3$, we have:

$$K_3^3 = \frac{1}{n}\sum_{i=1}^n \mathbb{E}\left|\frac{U_i^TU_i - E(U_i^TU_i)}{f}\right|^3$$

$$\leq \max_i \frac{\mathbb{E}(U_i^TU_i)^3 + (EU_i^TU_i)^3}{f^3} \leq 2\max_i \frac{\mathbb{E}(U_i^TU_i)^3}{\|\bar{\mathbb{V}}_n\|_F^3}$$

$$\leq 2\eta_n^3\lambda_1^3 \frac{\mathbb{E}\|V_\perp^T(X_iX_i^T-\Sigma)v_1\|^6}{\|\mathbb{M}\|_F^3}$$

Finally, for $J_1$ we have:

$$J_n = \frac{\sum_{i=1}^n \mathrm{Var}(U_i^TU_i)}{(nf)^2} \leq \frac{\sum_{i=1}^n \mathbb{E}(U_i^TU_i)^2}{n^2f^2}$$

$$\leq \frac{\eta_n^2\lambda_1^2}{n} \frac{\mathbb{E}[\|V_\perp(X_1X_1^T-\Sigma)v_1\|^4]}{\|\mathbb{M}\|_F^2}$$

The first makes $L_{3,2}$, $K_3^3/n$ and $J_n$ go to zero. The two conditions also imply $\frac{\mathbb{E}[\|V_\perp(X_1X_1^T-\Sigma)v_1\|^3]}{\|\mathbb{M}\|_F^3} = o(\sqrt{n})$, which implies $L_{3,1} \to 0$.

Finally, we collect remainder terms and show that their contribution to the inner product is negligible using anti-concentration. Observe that,

$$\sup_{t\in\mathbb{R}}\left|P(n/\eta_n \sin^2(w,v) \leq t) - P(Z_n^TZ_n \leq t)\right|$$

$$= \sup_{t\in\mathbb{R}}\left|P\left(R_1 \cdot \frac{(S_n+R_2+R_3)^T(S_n+R_2+R_3)}{f} \leq t\right) - P\left(\frac{Z_n^TZ_n}{f} \leq t\right)\right| \tag{S.11}$$

Now will will lower bound the above quantity. Observe that

$$P\left(R_1 \cdot \frac{(S_n+R_2+R_3)^T(S_n+R_2+R_3)}{f} \leq t\right)$$

$$\geq P\left(R_1 \cdot \frac{S_n^TS_n}{f}\left(1 + \frac{2\|R_2\| + 2\|R_3\|_2}{\sqrt{S_n^TS_n}}\right) + \frac{R_1 \cdot \|R_2+R_3\|^2}{f} \leq t\right) \tag{S.12}$$

$$= P\left(R' \cdot \frac{S_n^TS_n}{f} + \widetilde{R} \leq t\right), \text{ say.}$$

Now, for $\delta_n = o(\sqrt{f})$, we have that:

$$P\left(S_n^TS_n \leq \delta_n^2\right) \leq \sup_{t\in\mathbb{R}}\left|P(S_n^TS_n \leq t) - P(Z_n^TZ_n \leq t)\right| + P(Z_n^TZ_n \leq \delta_n^2) \to 0 \tag{S.13}$$

Note that $\delta_n = o(1)$ suffices since $f$ is bounded away from zero under Eq 8 as shown in Eq S.10.

Now, choose $\epsilon_n$ satisfying $\epsilon_n = o(1)$ $\epsilon_n = \omega\left(\sqrt{\frac{\eta_n^3 M_d^2 \log d}{n}}\right)$, define the set:

$$\mathcal{G} = \left\{ \left| R' - 1 \right| \leq \epsilon_n, \; \left| \widetilde{R} \right| \leq \epsilon_n \right\}$$

so that $P(\mathcal{G}^c) \to 0$ with the choice of $\delta_n$ in Eq. S.13. By using the fact that, for any two sets $A$ and $B$, $1 \geq P(A) + P(B) - P(A \cap B)$ and hence $P(A \cap B) \geq P(A) - P(B^c)$, we have that:

$$
\begin{aligned}
& P\left( R' \cdot \frac{S_n^T S_n}{f} + \widetilde{R} \leq t \right) \\
&= P\left( R' \cdot S_n^T S_n / f + \widetilde{R} \leq t \;\cap\; \mathcal{G} \right) + P\left( R' \cdot S_n^T S_n / f + \widetilde{R} \leq t \;\cap\; \mathcal{G}^c \right) \qquad (\text{S.14}) \\
&\geq P\left( \frac{S_n^T S_n}{f} \leq \frac{t}{1 + \epsilon_n} - \epsilon_n \right) - P(\mathcal{G}^c)
\end{aligned}
$$

Therefore,

$$
\begin{aligned}
& P\left( \frac{n/\eta_n \sin^2(w,v)}{f} \leq t \right) - P\left( \frac{Z_n^T Z_n}{f} \leq t \right) \\
&\geq P\left( \frac{S_n^T S_n}{f} \leq \frac{t}{1 + \epsilon_n} - \epsilon_n \right) - P\left( \frac{Z_n^T Z_n}{f} \leq \frac{t}{1 + \epsilon_n} - \epsilon_n \right) \qquad (\text{S.15}) \\
&\quad + P\left( \frac{Z_n^T Z_n}{f} \leq \frac{t}{1 + \epsilon_n} - \epsilon_n \right) - P\left( \frac{Z_n^T Z_n}{f} \leq t \right) - P(\mathcal{G}^c) = I + II - III
\end{aligned}
$$

Now, we may upper bound $III \to 0$ arising from our choice of $\delta_n$, and $II$ goes to 0 if the conditions of Proposition B.1 are satisfied, and $I \to 0$ due to Proposition B.3.

Now for the upper bound, since $\|R_i\|_2 \geq 0$, observe that we may bound Eq S.11 with:

$$
\begin{aligned}
& P\left( R_1 \cdot \frac{(S_n + R_2 + R_3)^T (S_n + R_2 + R_3)}{f} \leq t \right) \\
&\leq P\left( R_1 \cdot \frac{S_n^T S_n}{f}\left(1 - \frac{2\|R_2\| + 2\|R_3\|}{\sqrt{S_n^T S_n}}\right) - \frac{R_1 \cdot \|R_2\|\,\|R_3\|}{f} \leq t \right)
\end{aligned}
$$

We may now lower bound the negative terms and arrive at an identical expression to the lower bound. The result follows. $\qquad\square$

With the central limit theorem in hand, we are now ready to give the proof for Corollary 1.

*Proof of Corollary 1.* Observe that the approximating distribution $Z_n^T Z_n$ has expectation $\mathrm{trace}(\bar{\mathbb{V}}_n)$ and variance $f = \|\bar{\mathbb{V}}_n\|_F$. Therefore, for any $M > 0$, it follows that:

$$
\begin{aligned}
& P\left( \frac{n/\eta_n \sin^2(\hat{v}_1, v_1) - \mathrm{trace}(\bar{\mathbb{V}}_n)}{f} > M \right) \\
&\leq \sup_{t \in \mathbb{R}} \left| P\left( n/\eta_n \sin^2(\hat{v}_1, v_1) > t \right) - P\left( Z_n^T Z_n > t \right) \right| + P\left( \frac{Z_n^T Z_n - \mathrm{trace}(\bar{\mathbb{V}}_n)}{f} > M \right)
\end{aligned}
$$

The first term goes to zero under the conditions of Theorem 1. Chebychev's inequality implies that there exists $M > 0$ such that the latter probability can be made smaller than $\epsilon/2$ for any $\epsilon > 0$. Hence,

$$\frac{n/\eta_n \sin^2(\hat{v}_1, v_1) - \operatorname{trace}(\bar{\mathbb{V}}_n)}{f} = O_P(1).$$

Therefore, under the conditions in Theorem 1,

$$\sin^2(\hat{v}_1, v_1) = \frac{\eta_n}{n} \left[ \operatorname{trace}(\bar{\mathbb{V}}_n) + O_P \left( \|\bar{\mathbb{V}}_n\|_F \right) \right]$$

We now derive bounds for $\operatorname{trace}(\bar{\mathbb{V}}_n)$ and $\|\bar{\mathbb{V}}_n\|_F$. Let $\Lambda_\perp$ be a diagonal matrix with $\Lambda_\perp(i,i) = (1 + \eta_n \lambda_{i+1}/n)/(1 + \eta_n \lambda_1/n)$, $i = 1, \ldots, d-1$. Recall that:

$$\mathbb{M} := \mathbb{E}\left[ V_\perp^T (X_1^T v_1)^2 X_1 X_1^T V_\perp \right]. \tag{S.16}$$

$$\bar{\mathbb{V}}_n = \frac{\eta_n}{n} V_\perp \left( \sum_i \Lambda_\perp^{i-1} \mathbb{M} \Lambda_\perp^{i-1} \right) V_\perp^T$$

So now observe that,

$$\|\bar{\mathbb{V}}_n\|_F = \frac{\eta_n}{n} \left\| \sum_i \Lambda_\perp^{i-1} \mathbb{M} \Lambda_\perp^{i-1} \right\|_F$$

$$\operatorname{trace}(\bar{\mathbb{V}}_n) = \frac{\eta_n}{n} \operatorname{trace}\left( \sum_i \Lambda_\perp^{i-1} \mathbb{M} \Lambda_\perp^{i-1} \right)$$

A direct calculation shows that the $k, \ell^{th}$ entry of the sum $\sum_i \Lambda_\perp^{i-1} \mathbb{M} \Lambda_\perp^{i-1}$ is:

$$\sum_{i \geq 1} \left( \frac{1 + \eta_n \lambda_{k+1}/n}{1 + \eta_n \lambda_1/n} \right)^{i-1} \left( \frac{1 + \eta_n \lambda_{\ell+1}/n}{1 + \eta_n \lambda_1/n} \right)^{i-1} \mathbb{M}(k, \ell)$$

$$\leq \frac{n \mathbb{M}(k, \ell)}{\eta_n} \frac{(1 + \frac{\lambda_1 \eta_n}{n})^2}{2\lambda_1 - (\lambda_{k+1} + \lambda_{k+1}) + \frac{\eta_n}{n}(\lambda_1^2 - \lambda_k \lambda_l)} \tag{S.17}$$

$$\leq \frac{n}{\eta_n} \frac{C \mathbb{M}(k, \ell)}{\lambda_1 - \lambda_2}$$

for some $0 < C < \infty$.

Therefore, by Eq 7, we have

$$\operatorname{trace}(\bar{\mathbb{V}}_n) \leq C \frac{\operatorname{trace}(\mathbb{M})}{\lambda_1 - \lambda_2} \leq C \frac{M_d}{\lambda_1 - \lambda_2}$$

$$\|\bar{\mathbb{V}}_n\|_F \leq \frac{C \|\mathbb{M}\|_F}{\lambda_1 - \lambda_2} \leq C' \frac{M_d}{\lambda_1 - \lambda_2}$$

The last step is true since:

$$\operatorname{trace}(\mathbb{M}) = \operatorname{trace}(\mathbb{E}\left[ V_\perp^T (X_1^T v_1)^2 X_1 X_1^T V_\perp \right])$$

$$\begin{aligned}
&= \text{trace}(\mathbb{E}\left[V_\perp^T(X_1 X_1^T - \Sigma)v_1 v_1^T(X_1 X_1^T - \Sigma)V_\perp\right]) \\
&= \mathbb{E}\left(\text{trace}\left[V_\perp^T(X_1 X_1^T - \Sigma)v_1 v_1^T(X_1 X_1^T - \Sigma)V_\perp\right]\right) \\
&= \mathbb{E}\|V_\perp^T(X_1 X_1^T - \Sigma)v_1\|^2 \leq M_d
\end{aligned}$$

Similarly,

$$\begin{aligned}
\|\mathbb{M}\|_F &= \left\|\mathbb{E}\left[V_\perp^T(X_1^T v_1)^2 X_1 X_1^T V_\perp\right]\right\|_F \\
&= \left\|\mathbb{E}\left[V_\perp^T(X_1 X_1^T - \Sigma)v_1 v_1^T(X_1 X_1^T - \Sigma)V_\perp\right]\right\|_F \\
&\leq \mathbb{E}\|X_1 X_1^T - \Sigma\|_{op}^2 = M_d
\end{aligned}$$

where in the last line we used the fact that $\left\|xx^T\right\|_{op} = \left\|xx^T\right\|_F$ for $x \in \mathbb{R}^d$ since $xx^T$ is rank 1. $\quad\square$

## B.2   Adaptation of high-dimensional central limit theorem

Let $U_1, \ldots, U_n$, be independent random vectors in $\mathbb{R}^p$ such that $E(U_i) = 0$ and $\text{Var}(U_i) = \mathbb{V}_i$. Define a Gaussian analog of $Y_i$, denoted $Z_i$, which satisfies $E(Z_i) = 0$ and $\text{Var}(Z_i) = \mathbb{V}_i$. Furthermore, let $\bar{\mathbb{V}}_n = \frac{1}{n}\sum_{i=1}^n \mathbb{V}_i$, $g_i = \text{Var}(U_i^T U_i)$, $f_1 = \text{trace}(\bar{\mathbb{V}}_n)$, and $f = \left\|\bar{\mathbb{V}}_n\right\|_F$. For $0 < \delta \leq 1$, $q = 2 + \delta$, and $\beta \geq 2$ define the following quantities:

$$\begin{aligned}
L_q^U &= \frac{1}{n}\sum_{i=1}^n \frac{E(U_i^T\bar{\mathbb{V}}_n U_i)^{q/2}}{n^{\delta/2}f^q} + \frac{1}{\binom{n}{2}}\sum_{1 \leq i < j \leq n} \frac{E(|U_i^T U_j|^q)}{n^\delta f^q} \\
L_q^Z &= \frac{1}{n}\sum_{i=1}^n \frac{E(Z_i^T\bar{\mathbb{V}}_n Z_i)^{q/2}}{n^{\delta/2}f^q} \\
K_\beta^\beta &= \frac{1}{n}\sum_{i=1}^n E\left|\frac{U_i^T U_i - E(U_i^T U_i)}{f}\right|^\beta \\
J_n &= \frac{\sum_{i=1}^n g_i}{(nf)^2}
\end{aligned}$$

The following proposition is an adaptation of [8], which is stated for IID random variables, to independent but non-identically distributed random variables. While the changes are minor, we provide a proof below detailing the adaptation for completeness.

**Proposition B.1.** *Suppose that* $L_q^U \to 0$, $L_q^Z \to 0$, $J_n \to 0$, $n^{1-\beta}K_\beta^\beta \to 0$. *Then,*

$$\sup_{t \in \mathbb{R}} \left|P\left(n\bar{U}_n^T\bar{U}_n \leq t\right) - P\left(n\bar{Z}_n^T\bar{Z}_n \leq t\right)\right| \to 0$$

*Proof.* Since a Lindeberg argument is easier with diagonals removed, we will show that the removal of these terms is negligible. Observe that:

$$\begin{aligned}
&\sup_{t \in \mathbb{R}} \left|P(n\bar{U}_n^T\bar{U}_n \leq t) - P(n\bar{Z}_n^T\bar{Z}_n \leq t)\right| \\
&\leq \sup_{t' \in \mathbb{R}} \left|P\left(\frac{n\bar{U}_n^T\bar{U}_n - f_1}{f} \leq t'\right) - P\left(\frac{\sum_{i \neq j}U_i^T U_j}{nf} \leq t'\right)\right|
\end{aligned}$$

$$+ \sup_{t' \in \mathbb{R}} \left| P\left( \frac{\sum_{i \neq j} U_i^T U_j}{nf} \leq t' \right) - P\left( \frac{\sum_{i \neq j} Z_i^T Z_j}{nf} \leq t' \right) \right|$$

$$+ \sup_{t' \in \mathbb{R}} \left| P\left( \frac{\sum_{i \neq j} Z_i^T Z_j}{nf} \leq t' \right) - P\left( \frac{n \bar{Z}_n^T \bar{Z}_n - f_1}{f} \leq t' \right) \right|$$

$$= I + II + III, \text{ say.}$$

We will start by bounding $III$. First note that $\frac{1}{\sqrt{n}} \sum_{i=1}^n Z_i \sim \mathcal{N}(0, \bar{\mathbb{V}}_n)$. Let $\bar{\mathbb{V}}_n = Q^T D Q$ denote the eigendecomposition, with diagonal entries of $D$ given by $\lambda_1 \geq \ldots \geq \lambda_d$ and let $g \sim \mathcal{N}(0, \mathrm{I}_d)$. It follows that:

$$n \bar{Z}_n^T \bar{Z}_n \stackrel{d}{=} (Q D^{1/2} Q^T g)^T (Q D^{1/2} Q^T g)$$
$$\stackrel{d}{=} g^T D g$$

Notice that $V := g^T D g \sim \sum_{r=1}^d \lambda_r \eta_r$, where $\eta_1, \ldots, \eta_d \sim \chi^2(1)$. Now define $R_n^Z = \frac{\frac{1}{n} \sum_{i=1}^n Z_i^T Z_i - f_1}{f}$. Notice that:

$$P\left( \frac{n \bar{Z}_n^T \bar{Z}_n - f_1}{f} \leq t \right) - P\left( \frac{\sum_{i \neq j} Z_i^T Z_j}{f} \leq t \right)$$

$$= P\left( \frac{n \bar{Z}_n^T \bar{Z}_n - f_1}{f} \leq t \right) - P\left( \frac{n \bar{Z}_n^T \bar{Z}_n - f_1}{f} - R_n^Z \leq t \right) \qquad \text{(S.18)}$$

$$\leq P(t' \leq V \leq t' + h_n) + P(|R_n^Z| > h_n)$$

Under the conditions $J_n \to 0$, $n^{1-\beta} K_\beta^\beta \to 0$, Nagaev's inequality implies that one may choose $h_n \to 0$ such that $P(|R_n^Z| > h_n) \to 0$. The desired anti-concentration for the first term in the previous display follows from Lemma S2 of [8]. We may also derive the lower bound $P(t' \leq V \leq t' + h_n) - P(|R_n^Z| > h_n)$ in a similar manner.

To adapt $II$, consider the smoothed indicator function:

$$g_{\psi,t}(x) = \left[ 1 - \min\{1, \max(x - t, 0)\}^4 \right]^4.$$

This function satisfies:

$$\max_{x,t} \{ |g'_{\psi,t}(x)| + |g''_{\psi,t}(x)| + |g'''_{\psi,t}(x)| \} < \infty$$

$$\mathbb{1}_{x \leq t} \leq g_{\psi,t} \leq \mathbb{1}_{x \leq t + \psi^{-1}}.$$

Therefore, we may bound the approximation error with smoothed indicator function by again using anti-concentration of the weighted $\chi^2$. In what follows, let:

$$S_n^U = \frac{1}{nf} \sum_{i \neq j} U_i^T U_j, \quad S_n^Z = \frac{1}{nf} \sum_{i \neq j} Z_i^T Z_j$$

We have that:

$$P(S_n^U \leq t) - P(S_n^Z \leq t)$$
$$\leq P(S_n^U \leq t) - P(S_n^Z \leq t + \psi^{-1}) + P(S_n^Z \leq t + \psi^{-1}) - P(S_n^Z \leq t)$$
$$\leq E g_{\psi,t}(S_n^U) - E g_{\psi,t}(S_n^Z) + III + P(t \leq V \leq t + \psi^{-1}).$$

An analogous argument establishes a lower bound of $g_{\psi,t}(S_n^U) - Eg_{\psi,t}(S_n^Z) - III - P(t - \psi^{-1} \leq V \leq t)$. Choosing $\psi_n \to \infty$, the last term goes to zero. A Lindeberg telescoping sum argument leads to the following bound for the leading term:

$$\left| Eg_{\psi,t}(S_n^U) - Eg_{\psi,t}(S_n^Z) \right| \leq \sum_{i=1}^{n} c_q (E|\Delta_i|^q + E|\Gamma_i|^q),$$

where:

$$H_i = \sum_{j=1}^{i=1} U_i + \sum_{j=i+1}^{n} Z_i, \quad \Delta_i = \frac{U_i^T H_i}{nf}, \quad \Gamma_i = \frac{Z_i^T H_i}{nf}.$$

We may use analogous reasoning to bound these terms. Let $\xi \sim N(0,1)$. Conditioning on $U_1 = u_i$, by Rosenthal's inequality:

$$
\begin{aligned}
\mathbb{E}\left[ |\Delta_i|^q \mid U_i \right] &\leq \sum_{j=1}^{i-1} \frac{\mathbb{E}[|U_j^T u_i|^q]}{n^q f^q} + \sum_{j=i+1}^{n} \frac{\mathbb{E}[|Z_j^T u_i|^q]}{n^q f^q} + n^{q/2} \frac{\left(u_i^T \bar{\mathbb{V}}_n u_i\right)^{q/2}}{n^q f^q} \\
&\leq \sum_{j=1}^{i-1} \frac{\mathbb{E}[|U_j^T u_i|^q]}{n^q f^q} + \sum_{j=i+1}^{n} \|\xi\|_q^q \frac{\left(u_i^T \bar{\mathbb{V}}_j u_i\right)^{q/2}}{n^q f^q} + \frac{\left(u_i^T \bar{\mathbb{V}}_n u_i\right)^{q/2}}{n^{q/2} f^q}
\end{aligned}
$$

(S.19)

Taking expectations, it follows that:

$$\sum_{i=1}^{n} \mathbb{E}\left[ |\Delta_i|^q \right] \lesssim \frac{1}{\binom{n}{2}} \sum_{1 \leq i < j \leq n} \frac{\mathbb{E}\left[ |U_i^T U_j|^q \right]}{n^\delta f^q} + \frac{1}{n} \sum_{i=1}^{n} \frac{\mathbb{E}\left| U_i^T \bar{\mathbb{V}}_n U_i \right|^{q/2}}{n^{\delta/2} f^q}$$

Now, for $\Gamma_i$, we may use Rosenthal's inequality so that:

$$\sum_{i=1}^{n} \mathbb{E}\left[ |\Gamma_i|^q \right] \leq \frac{1}{n} \sum_{i=1}^{n} \frac{\mathbb{E}\left| U_i^T \bar{\mathbb{V}}_n U_i \right|^{q/2}}{n^\delta \delta f^q} + \frac{1}{n} \sum_{i=1}^{n} \frac{\mathbb{E}\left[ |Z_i^T \bar{\mathbb{V}}_n Z_i|^{q/2} \right]}{n^\delta \delta f^q} + \frac{1}{n} \sum_{i=1}^{n} \frac{\mathbb{E}\left( Z_i^T \bar{\mathbb{V}}_n Z_i \right)^{q/2}}{n^{q/2} f^q}$$

While omitted in the original proof, in the IID case, the latter terms may be bounded by using an eigendecomposition along with properties of the Gaussian. However, since the $Z_i$ do not have variance matrix $\mathbb{V}_n$, we instead oppose the additional condition for $L_q^Z$. By the assumptions made in theorem, it follows that $II \to 0$.

Finally, for $I$, we have that:

$$
\begin{aligned}
&P\left( \frac{n\bar{U}_n^T \bar{U}_n - f_1}{f} \leq t \right) - P\left( \frac{\sum_{i \neq j} U_i^T U_j}{nf} \leq t \right) \\
&\leq P(S_n^X \leq t + h_n) - P(S_n^U \leq t + h_n) + P(|R_n^X| > h_n) \\
&\quad + P(t \leq V \leq t + 2h_n) + P(|S_n^Z| > h_n)
\end{aligned}
$$

Using bounds from $II$ and $III$ along with anti-concentration properties, we may conclude that $I \to 0$.

$\square$

## B.3 Supporting lemmas for CLT

In several of our lemmas, we use the following technique from [4] that facilitates analysis for initializations from a uniform distribution on $\mathcal{S}^{d-1}$ particularly when $d$ is large.

**Proposition B.2** (Trace trick). *Suppose that $u$ is drawn from a uniform distribution on $\mathcal{S}^{d-1}$. Then, for any $A \in \mathbb{R}^{d \times d}$ and $v \in \mathbb{R}^d$ satisfying $\|v\| = 1$, with probability at least $1 - C\delta$, for some $C > 0$ independent of $A$ and $0 < \delta < 1$,*

$$\frac{u^T A^T A u}{(v^T u)^2} \leq \frac{\log(1/\delta) \, \text{trace}(AA^T)}{\delta^2}$$

*Proof.* First, we recall the well-known fact that $u = g/\|g\|$, where $g \sim N(0, I_d)$. Therefore, $\|g\|$ cancels as follows:

$$\frac{u^T A^T A u}{(v^T u)^2} = \frac{g^T A^T A g}{(v^T g)^2}$$

Furthermore, observe that $g^T A^T A g$ may be viewed as a weighted sum of independent $\chi^2(1)$ random variables. In particular, by an eigendecomposition argument, for $\eta_1, \ldots \eta_r \sim \chi^2(1)$ and $A = VDV^T$,

$$g^T(VDV^T)(VDV^T)g = g^T V D^2 V^T g$$
$$\overset{d}{=} g^T D^2 g$$
$$= \sum_{r=1}^{p} \lambda_r^2 \eta_r = \psi, \text{ say}$$

where above we used the fact that $V^T g \sim N(0, I_d)$. Now observe that $\mathbb{E}[\psi] = \sum_{r=1}^{p} \lambda_r^2 = \|A\|_F^2$ and that $\eta_r$ is sub-Exponential. Therefore, by by Bernstein's inequality (see for example Theorem 2.8.2 of [7]), for some $K > 0$, $C_1 > 0$, $0 < \delta < 1$,

$$P\left(\psi - \mathbb{E}[\psi] > (\log(1/\delta) - 1)\|A\|_F^2\right) \leq \exp\left\{-\min\left(\frac{\log^2(1/\delta)\|A\|_{\mathcal{S}_2}^4}{4K^2\|A\|_{\mathcal{S}_4}^4}, \frac{\log(1/\delta)\|A\|_{\mathcal{S}_2}^2}{2K\|A\|_{\mathcal{S}_\infty}^2}\right)\right\}$$
$$\leq \exp\left\{-\min\left(\frac{\log^2(1/\delta)}{4K^2}, \frac{\log(1/\delta)}{2K}\right)\right\} \leq C_1\delta$$

where above $\|\cdot\|_{\mathcal{S}_p}$ is the $p$th Schatten-Norm, defined as $(\sum_{r=1}^{d} s_r^p)^{1/p}$, where $s_r$ is the $r$th singular value and satisfies $\|\cdot\|_{\mathcal{S}_q} \leq \|\cdot\|_{\mathcal{S}_p}$ for $p \leq q$. Now for the denominator, since $v^T g \sim N(0,1)$ and $(v^T g)^2 \sim \chi^2(1)$, Proposition B.3 yields:

$$P((v^T g)^2 \leq \delta^2) \leq \frac{2\delta}{\sqrt{\pi}}$$

The result follows. $\qquad\square$

The following anti-concentration result for weighted $\chi^2$ distributions is also used in several places.

**Proposition B.3** (Weighted $\chi^2$ anti-concentration, [8]). *Let $a_1 \geq \cdots \geq a_p \geq 0$ such that $\sum_{r=1}^{p} a_i^2 = 1$ and suppose that $\xi_1, \ldots, \xi_p \sim \chi^2(1)$. Then,*

$$\sup_{t \in \mathbb{R}} P\left(t \leq \sum_{r=1}^{p} a_r \xi_r \leq t + h\right) \leq \sqrt{\frac{4h}{\pi}}$$

We now present a concentration result for matrix products that follow immediately from Corollary 5.4 of [3].

**Lemma B.1** (Expectation bounds for operator norms of of matrix products). *Let $\mathcal{B}_k = \prod_{j=1}^{k}(I + \eta_n X_j X_j^T / n)$. We have,*

$$\mathbb{E}\|\mathcal{B}_k - \mathbb{E}\mathcal{B}_k\|^2 \leq \frac{M_d e \eta_n^2 (1 + 2\log d)k}{n^2}(1 + \eta_n \lambda_1/n)^{2k}. \tag{S.20}$$

*For the expectation, we have, if $\frac{(1 + 2\log d) M_d \eta_n^2}{n} \leq 1$:*

$$\mathbb{E}\|\mathcal{B}_k\|^2 \leq \exp\left(2\sqrt{2M_d \frac{k\eta_n^2}{n^2}\left(2M_d \frac{k\eta_n^2}{n^2} \vee \log d\right)}\right)(1 + \eta_n \lambda_1/n)^{2k}. \tag{S.21}$$

*Proof.* We invoke Corollary 5.4 in [3] with $\|\mathbb{E}(I + \eta_n/n X_i X_i^T)\| \leq 1 + \eta_n \lambda_1/n$, $\sigma_i^2 = M_d \frac{\eta_n^2}{n^2}$, and $\nu = M_d \frac{k\eta_n^2}{n^2}$. Note that for a random matrix $M$ with Schatten norm $\|M\|_{\mathcal{S}_p}$, $\mathbb{E}\|M\| \leq \sqrt{\mathbb{E}\|M\|_{\mathcal{S}_p}^2}$ and hence the same argument as in their proof invoking Eq 5.5 and 5.6 works. $\square$

**Lemma B.2** (Concentration of the norm for the CLT). *For some $C > 0$, and any $\epsilon > 0$, $0 < \delta < 1$,*

$$P\left(\left|\frac{\|B_n u_0\|}{|a_1|(1 + \eta_n \lambda_1/n)^n} - 1\right| \geq \epsilon\right)$$

$$\leq \frac{d\exp\left(-\eta_n(\lambda_1 - \lambda_2) + \frac{\eta_n^2}{n}(\lambda_1^2 + M_d)\right) + \frac{\eta_n^2}{n}M_d \exp\left(\frac{\eta_n^2}{n}\right)}{4\log^{-1}(1/\delta)\delta^2\epsilon^2\left(1 + \frac{\eta_n^2 \lambda_1^2}{n}\right)} + \frac{e^2\eta_n^2 M_d(1 + \log d)}{n\epsilon^2} + C\delta$$

*Proof.* Consider the bound:

$$\left|\frac{\|B_n u_0\|}{|a_1|(1 + \eta_n \lambda_1/n)^n} - 1\right| \leq \left|\frac{\|B_n v_1 a_1\| - \|a_1 T_0 v_1\|}{|a_1|(1 + \eta_n \lambda_1/n)^n}\right| + \frac{\|B_n V_\perp (V_\perp^T u_0)\|}{|a_1|(1 + \eta_n \lambda_1/n)^n}$$

We will start by bounding the second term.

Using Proposition B.2, observe that, with probability at least $1 - C\delta$,

$$\frac{\|(B_n V_\perp V_\perp^T g)\|^2}{|v_1^T g|^2 (1 + \eta_n \lambda_1/n)^{2n}} \leq \frac{\log(1/\delta)\mathrm{trace}(V_\perp B_n B_n V_\perp^T))}{\delta^2 (1 + \eta_n \lambda_1/n)^{2n}}$$

Let $\mathcal{G}$ denote the good set for which the upper bound above holds. Markov's inequality on the good set, together with Lemma 5.2 of [4] with $\mathcal{V}_n \leq M_d$ yields that:

$$P\left(\frac{\|B_n V_\perp V_\perp^T g\|}{(1 + \eta_n \lambda_1/n)^n} \geq \epsilon/2 \cap \mathcal{G}\right)$$

$$\leq \frac{d \exp\left(-\eta_n(\lambda_1 - \lambda_2) + \frac{\eta_n^2}{n}(\lambda_1^2 + M_d)\right) + \frac{\eta_n^2}{n} M_d \exp\left(\frac{\eta_n^2}{n}\right)}{4\delta^2 \log^{-1}(1/\delta) \, \epsilon^2 \left(1 + \frac{\eta_n^2 \lambda_1^2}{n}\right)}$$

Now we will bound the first summand. By Lemma B.1 Eq S.20, we have by Markov's inequality,

$$P\left(\frac{\|(B_n - T_0)\|_{op}}{(1 + \eta_n \lambda_1/n)^n} > \epsilon/2\right) \leq \frac{e^2 M_d(1 + \log d)}{n\epsilon^2}$$

Combining the two bounds and the probability of $\mathcal{G}^c$, the result follows.

$\square$

**Lemma B.3** (Negligibility of $V_\perp$ for the CLT). *Let $V_\perp$ denote the matrix of eigenvectors orthogonal to $v_1$. Also let $\lambda_i$ denote the $i^{th}$ largest eigenvalue of $\Sigma$. For some $C > 0$, and any $\epsilon > 0$, $0 < \delta < 1$,*

$$P\left(\sqrt{\frac{n}{\eta_n}} \, \frac{\|V_\perp V_\perp^T B_n V_\perp V_\perp^T u_0\|}{|a_1|(1 + \frac{\eta_n \lambda_1}{n})^n} \geq \epsilon\right)$$

$$\leq \frac{nd \log(1/\delta) \exp\{-2\eta_n(\lambda_1 - \lambda_2) + \eta_n^2(\lambda_1^2 + M_d)/n\}}{\eta_n \epsilon^2 \delta^2} + \frac{eM_d^2(1 + 2\log d)\eta_n^2 \epsilon^{-2} \log(1/\delta)\delta^{-2}}{n2(\lambda_1 - \lambda_2) + \eta_n^2(\lambda_1^2 - \lambda_2^2 - M_d)} + C\delta$$

*Proof.* We consider bounding the squared quantity. We have, with probability at least $1 - C\delta$, using Proposition B.2, this quantity is upper bounded by:

$$\frac{\left\|(V_\perp V_\perp^T B_n V_\perp V_\perp)g\right\|^2}{(v_1^T g)^2(1 + \eta_n \lambda_1/n)^{2n}}$$

$$\leq \frac{\text{trace}\left((V_\perp V_\perp^T B_n V_\perp V_\perp^T)(V_\perp V_\perp^T B_n V_\perp V_\perp^T)^T\right)}{\delta_n(v_1^T g)^2(1 + \eta_n \lambda_1/n)^{2n}}$$

$$= \frac{\text{trace}\left(V_\perp^T B_n V_\perp V_\perp^T B_n V_\perp\right)}{\delta_n^3(1 + \eta_n \lambda_1/n)^{2n}}$$

Now we will bound the expectation of the numerator.

We will denote $\eta = \frac{\eta_n}{n}$ for simplicity. Let $U_i = I + \eta X_i X_i^T$ and $Y_i = X_i X_i^T - \Sigma$. We have that:

$$\alpha_n := \mathbb{E}\left\langle B_n V_\perp V_\perp^T B_n^T, V_\perp V_\perp^T\right\rangle$$

$$= \mathbb{E}\left\langle B_{n-1} V_\perp V_\perp^T B_{n-1}^T, U_n V_\perp V_\perp^T U_n^T\right\rangle$$

$$= \left\langle \mathbb{E}B_{n-1} V_\perp V_\perp^T B_{n-1}^T, \mathbb{E}U_n V_\perp V_\perp^T U_n^T\right\rangle \tag{S.22}$$

Now we have:

$$\mathbb{E}U_n V_\perp V_\perp^T U_n^T = \mathbb{E}(I + \eta\Sigma) V_\perp V_\perp^T (I + \eta\Sigma)^T + \eta^2 \mathbb{E}Y_n V_\perp V_\perp^T Y_n^T$$

$$\preceq (1 + 2\eta\lambda_2 + \lambda_2^2 \eta^2)V_\perp V_\perp^T + \eta^2 M_d(V_\perp V_\perp^T + v_1 v_1^T)$$

$$\preceq (1 + 2\eta\lambda_2 + \lambda_2^2 \eta^2 + \eta^2 M_d^2)V_\perp V_\perp^T + \eta^2 M_d v_1 v_1^T \tag{S.23}$$

Finally, using Eqs S.22 and S.23, we have:

$$\alpha_n \leq \left(1 + 2\eta\lambda_2 + \eta^2(\lambda_2^2 + M_d)\right)\alpha_{n-1} + \eta^2 M_d \left\langle \mathbb{E}B_{n-1} V_\perp V_\perp^T B_{n-1}^T, v_1 v_1^T\right\rangle \tag{S.24}$$

We will use the fact that,

$$\langle (I + \eta\Sigma)^{n-1} V_\perp V_\perp^T (I + \eta\Sigma)^{n-1}, v_1 v_1^T \rangle = 0.$$

Thus, for some $N$ such that the condition $\eta_n^2 M_d (1 + 2\log d)/n \leq 1$ holds for all rows of the triangular array with index $n > N$, we have by Lemma B.1,

$$
\begin{aligned}
&\langle \mathbb{E} B_{n-1} V_\perp V_\perp^T B_{n-1}^T, v_1 v_1^T \rangle \\
&= \langle \mathbb{E}(B_{n-1} - (I + \eta\Sigma)^{n-1}) V_\perp V_\perp^T (B_{n-1} - (I + \eta\Sigma)^{n-1})^T, v_1 v_1^T \rangle \\
&\leq \| \mathbb{E}(B_{n-1} - (I + \eta\Sigma)^{n-1}) V_\perp V_\perp^T (B_{n-1} - (I + \eta\Sigma)^{n-1})^T \| \\
&\leq \mathbb{E} \| B_{n-1} - (I + \eta\Sigma)^{n-1} \|^2 \\
&\leq M_d e \eta^2 n (1 + 2\log d)(1 + \eta_n \lambda_1/n)^{2(n-1)}.
\end{aligned}
$$

Thus, Eq S.24 gives:

$$
\begin{aligned}
\alpha_n &\leq \underbrace{\left(1 + 2\eta\lambda_2 + \eta^2(\lambda_2^2 + M_d)\right)}_{c_1} \alpha_{n-1} + \eta^4 M_d^2 e(1 + 2\log d) \underbrace{(n-1)(1 + \eta\lambda_1)^{2(n-1)}}_{(n-1)c_2^{n-1}} \\
&= c_1 \alpha_{n-1} + \eta^4 M_d^2 e(1 + 2\log d)(n-1)c_2^{n-1} \\
&= c_1^n \alpha_0 + \eta^4 M_d^2 e(1 + 2\log d) \sum_i c_1^{i-1}(n-i)c_2^{n-i} \\
&\leq c_2^n \left( d(c_1/c_2)^n + \frac{e M_d^2 (1 + 2\log d)\eta^4 n}{c_2 - c_1} \right) \\
&\leq (1 + \eta_n \lambda_1/n)^{2n} \Big( d(1 - \lambda_1^2 \eta_n^2/n) \exp\{-2\eta_n(\lambda_1 - \lambda_2) + \eta_n^2(\lambda_1^2 + M_d)/n\} \\
&\qquad + \frac{e M_d^2(1 + 2\log d)\eta_n^3/n^2}{2(\lambda_1 - \lambda_2) + \eta_n^2/n(\lambda_1^2 - \lambda_2^2 - M_d)} \Big)
\end{aligned}
$$

where above we used the fact $e^x(1 - \frac{x^2}{n}) \leq (1 + \frac{x}{n})^n \leq e^x$ for $|x| \leq n$ to bound $(c_1/c_n)^n$. $\qquad\square$

**Lemma B.4** (Negligibility of higher-order Hoeffding projections for the CLT). *Let $\beta_n = \frac{\eta_n^2 M_d}{n}$ and suppose that $0 \leq \beta_n \leq 1$. Then, for some $C > 0$ and any $\epsilon > 0$,*

$$P\left( \frac{\sqrt{\frac{n}{\eta_n}} \left\| V_\perp V_\perp^T \sum_{k>1} T_k v_1 \right\|}{(1 + \frac{\eta_n \lambda_1}{n})^n} > \epsilon \right) \leq \frac{C\beta_n \eta_n}{(1 - \beta_n)\epsilon^2}$$

*Proof.* By Markov's inequality, it follows that:

$$P\left( \frac{\frac{\sqrt{n}}{\eta_n} \left\| V_\perp V_\perp^T \sum_{k>1} T_k v_1 \right\|}{(1 + \frac{\eta_n \lambda_1}{n})^n} > \epsilon \right) \leq \frac{\frac{n}{\eta_n^2} \mathbb{E}\left[ \left\| V_\perp V_\perp^T \sum_{k>1} T_k v_1 \right\|^2 \right]}{\epsilon^2 (1 + \frac{\eta_n \lambda_1}{n})^{2n}}$$

Now, by submultiplicativity of the operator norm and the fact that $\mathbb{E}[(P_{S_1} T)^T (P_{S_2})T] = 0$ for any two Hayek projections, the numerator is upper bounded by:

$$\left(\frac{n}{\eta_n}\right)\sum_{k=2}^{n}\left(\frac{\eta_n}{n}\right)^{2k}\sum_{|S|=k}\mathbb{E}\left[(v'A_S u_0)^2\right] \leq \left(\frac{n}{\eta_n}\right)\sum_{k=2}^{n}\sum_{|S|=k}\left(\frac{\eta_n}{n}\right)^{2k}\mathbb{E}\left[\|A_S\|_{op}^2\right]$$

$$\leq \left(\frac{n}{\eta_n}\right)\sum_{k=2}^{n}\left(\frac{\eta_n}{n}\right)^{2k}\sum_{|S|=k}\left(1+\frac{\eta_n\lambda_1}{n}\right)^{2(n-k)}M_d^k$$

$$\leq \eta_n M_d\left(1+\frac{\eta_n\lambda_1}{n}\right)^{2n}\sum_{k=2}^{n}\left(\frac{M_d\eta_n^2}{n}\right)^{k-1}$$

$$\leq \left(1+\frac{\eta_n\lambda_1}{n}\right)^{2n}\frac{\beta_n\eta_n M_d}{1-\beta_n}$$

The result follows.

$\square$

# C   Consistency of the online bootstrap

In this section, we provide the detailed proof of Bootstrap consistency, i.e Theorem 2.

## C.1   Proof of bootstrap consistency

*Proof of Theorem 2.* Similar to the CLT, we will establish the negligibility of remainder terms and then use anti-concentration terms to argue that the contribution to the Kolmogorov distance is small. We then show that the bootstrap covariance of the main term approaches the weighted $\chi^2$ approximation in Theorem 1 with high probability. Let $\widehat{v}_1$ denote the leading eigenvector estimated from Oja's algorithm and let $\widehat{V}_\perp$ denote its orthogonal complement. Again, we have that:

$$\frac{n}{\eta_n}\sin^2(v_1^*,\hat{v}_1) = \frac{n}{\eta_n}\frac{(B_n^* u_0)^T\widehat{V}_\perp\widehat{V}_\perp^T(B_n^* u_0)}{\|B_n^* u_0\|^2}$$

$$= \frac{(\sqrt{n/\eta_n}\widehat{V}_\perp\widehat{V}_\perp^T B_n^* u_0)^T(\sqrt{n/\eta_n}\widehat{V}_\perp\widehat{V}_\perp^T B_n^* u_0)}{\|B_n^* u_0\|^2}$$

We aim to show that the bootstrap distribution conditional on the data is close to the weighted $\chi^2$ approximation with high probability; therefore we may work the good set $\mathcal{A}_n$. With the a slight abuse of notation, in the remainder terms below, $O_P$ will be on the measure restricted to $\mathcal{A}_n$.

We first approximate the norm using Lemma C.3. Analogous to the CLT, the corresponding remainder is given by:

$$R_1^* = \frac{\|B_n^* u_0\|^2}{a_1^2(1+\frac{\eta_n}{n}\lambda_1)^{2n}} = 1 - O_P\left(\sqrt{d}\exp\left(-\frac{\eta_n}{2}(\lambda_1-\lambda_2)\right)+\sqrt{\frac{\eta_n^2 M_d\log d}{n}}+\frac{\eta_n\alpha_n}{\sqrt{n}}\right)$$

Next, we bound the contribution of the higher-order Hoeffding projections. This step is different from the CLT in the sense that we handle both $v_1$ and $V_\perp$, using the fact that on the good set, even the Frobenius norm of certain terms are well-behaved. By Lemma C.4 we have that:

$$R_3^* := \sqrt{\frac{n}{\eta_n}}\cdot\frac{\widehat{V}_\perp\widehat{V}_\perp^T(B_n^*-T_1^*)u_0}{|a_1|(1+\eta_n/n\lambda_1)^n} = O_P\left(\exp\left(\sqrt{\frac{CM_d^2\eta_n^2\log d}{n}}\right)\sqrt{\frac{\alpha_n^4\eta_n^3}{n}}\right)$$

Next, we bound the contribution of $V_\perp$ to the Hájek projection using Lemma C.6, as long as $\lambda_1 M_d (\log d)^2 \frac{\eta_n^2}{n} \to 0$,

$$R_2^* = \sqrt{\frac{n}{\eta_n}} \cdot \frac{\widehat{V}_\perp \widehat{V}_\perp^T T_1^* V_\perp V_\perp^T u_0}{|a_1|(1 + \eta_n/n\lambda_1)^n} = O_P\left(\sqrt{\frac{\alpha_n M_d \eta_n^2}{n(\lambda_1 - \lambda_2)}}\right)$$

The final remainder term arises from the disparity between the orthogonal complements and the residuals of matrix products from their expectation. By Lemma C.2, with $\Delta_i = X_i X_i^T - X_{i-1} X_{i-1}^T$,

$$R_4^* = \sqrt{\frac{n}{\eta_n}} \left\| \frac{\widehat{V}_\perp \widehat{V}_\perp^T T_1^* v_1 (v_1^T u_0)}{|v_1^T u_0|(1 + \eta_n \lambda_1/n)^n} - \frac{\eta_n}{n} \sum_i W_i D_{i-1} \Delta_i v_1 \right\| = O_P\left(\sqrt{\frac{M_d \alpha_n \eta_n^3 \log d}{n}}\right)$$

Now, define:

$$S_n^* = \sqrt{\frac{n}{\eta_n}} \frac{V_\perp V_\perp^T T_1^* v_1}{(1 + \frac{\eta_n \lambda_1}{n})^n}$$

Consider the following bound:

$$P\left\{ \sup_{t \in \mathbb{R}} \left| P^*(n/\eta_n \sin^2(v_1^*, \widehat{v}_1) \le t) - P(Z^T Z \le t) \right| > \epsilon \right\}$$

$$= P_{\mathcal{A}_n}\left\{ \sup_{t \in \mathbb{R}} \left| P^*\left( R_1^* \cdot \frac{(S_n^* + R_2^* + R_3^* + R_4^*)^T (S_n^* + R_2^* + R_3^* + R_4^*)}{f} \le t \right) - P\left( \frac{Z^T Z}{f} \le t \right) \right| > \epsilon \right\}$$

$$+ P_{\mathcal{A}_n^c}\left\{ \sup_{t \in \mathbb{R}} \left| P^*\left( R_1^* \cdot \frac{(S_n^* + R_2^* + R_3^* + R_4^*)^T (S_n^* + R_2^* + R_3^* + R_4^*)}{f} \le t \right) - P\left( \frac{Z^T Z}{f} \le t \right) \right| > \epsilon \right\}$$
(S.25)

The second term is easily upper-bounded by $P(\mathcal{A}_n^c) \to 0$, so we will bound the first term. To lower bound the Kolmogorov metric, we may follow the same reasoning used in Eqs S.12, S.14, S.15, to deduce, on the good set $\mathcal{A}_n$, we have the lower bound:

$$P^*\left( \frac{S_n^{*T} S_n^*}{f} \le \frac{t}{1 + \epsilon_n} - \epsilon_n \right) - P\left( \frac{Z^T Z}{f} \le \frac{t}{1 + \epsilon_n} - \epsilon_n \right)$$

$$+ P\left( \frac{Z^T Z}{f} \le \frac{t}{1 + \epsilon_n} - \epsilon_n \right) - P\left( \frac{Z^T Z}{f} \le t \right) - P^*(G_{boot} \cap \mathcal{A}_n) = I^* + II^* + III^*$$

where $G_{boot}$ satisfies $P(G_{boot}^c) = 0$ and for some $\epsilon_n \to 0$, is defined as:

$$G_{boot} = \{|R_1^* - 1| \le \epsilon_n, |R_2^*|, |R_3^*|, |R_4^*| \le \epsilon_n \}$$

For $I$, we may use Lemma 1, which establishes that bootstrap version of the covariance matrix, which consists of empirical covariances, is close to the Gaussian approximation, implying, by our Gaussian comparison result Lemma C.1:

$$I^* = O_P\left( \left( \frac{\mathbb{E}[\|X_i X_i^T - \Sigma\|^4]}{n(\lambda_1 - \lambda_2)\|M\|_F^2} \right)^{1/4} \right)$$

For $II^*$, we may use the anti-concentration result and $P^*(G_{boot} \cap \mathcal{A}_n) \xrightarrow{P} 0$ by Markov's inequality since the Lemmas hold for the unconditional measure, which is the expectation of the bootstrap measure. We may use analogous reasoning to the CLT for the upper bound and the result follows. $\square$

## C.2 Proof of Lemma 1

*Proof.* Let $Y_i := X_i X_i^T - \Sigma$. Also let $M_i = \mathbb{E}[D_{i-1} Y_i v_1 v_1^T Y_i D_{i-1}]$. First note that

$$
\mathbb{E}^* Z Z^T - \bar{\mathbb{V}}_n = \frac{\eta_n}{2n} \sum_i D_{i-1}(Y_i - Y_{i-1}) v_1 v_1^T (Y_i - Y_{i-1}) D_{i-1}
$$

$$
= \frac{\eta_n}{n} \sum_i \frac{\left(D_{i-1} Y_i v_1 v_1^T Y_i D_{i-1} - M_i\right) + \left(D_{i-1} Y_{i-1} v_1 v_1^T Y_{i-1} D_{i-1} - M_i\right)}{2}
$$

$$
+ \frac{\eta_n}{n} \sum_i \left(D_{i-1} Y_i v_1 v_1^T Y_{i-1} D_{i-1} + D_{i-1} Y_{i-1} v_1 v_1^T Y_i D_{i-1}\right) \qquad \text{(S.26)}
$$

We first compute trace.

$$
\text{trace}(\mathbb{E}^* Z Z^T - \bar{\mathbb{V}}_n) = \frac{\eta_n}{2n} \sum_i \underbrace{\left(\|D_{i-1} Y_i v_1\|^2 - \mathbb{E}\|D_{i-1} Y_i v_1\|^2\right)}_{U_{1,i}}
$$

$$
+ \frac{\eta_n}{2n} \sum_i \underbrace{\left(\|D_{i-1} Y_{i-1} v_1\|^2 - \mathbb{E}\|D_{i-1} Y_i\|^2\right)}_{U_{2,i}}
$$

$$
+ \frac{\eta_n}{n} \sum_i \underbrace{v_1 Y_i D_{2(i-1)} Y_{i-1} v_1}_{U_{3,i}}
$$

The last step is true because $D_{i-1}^2 = D_{2(i-1)}$. We start with the first term.

$$
\mathbb{E} U_{i,1}^2 \leq \mathbb{E}\|D_{i-1} Y_i v_1\|^4 \leq \mathbb{E}\|Y_i\|^4 \left(\frac{1 + \eta_n \lambda_2/n}{1 + \eta_n \lambda_1/n}\right)^{4(i-1)}
$$

$$
\text{Var}\left(\sum_i U_{1,i}\right) \leq \mathbb{E}\|Y_1\|^4 \sum_i \left(\frac{1 + \eta_n \lambda_2/n}{1 + \eta_n \lambda_1/n}\right)^{4(i-1)}
$$

$$
\leq \frac{n}{\eta_n(\lambda_1 - \lambda_2)}
$$

$$
\leq \frac{n}{\eta_n} \mathbb{E}\|Y_1\|^4 \min\left(\frac{1}{\lambda_1 - \lambda_2}, \eta_n\right)
$$

Finally,

$$
\mathbb{E}[U_{3,i}^2] \leq \mathbb{E}\left(v_1 Y_i D_{2(i-1)} Y_{i-1} v_1\right)^2 \leq M_d^2 \left(\frac{1 + \eta_n \lambda_2/n}{1 + \eta_n \lambda_1/n}\right)^{2(i-1)}
$$

Thus, we have

$$
\frac{\eta_n}{2n} \sum_i U_{1,i} = O_P\left(\sqrt{\frac{\mathbb{E}\|Y_1\|^4}{n(\lambda_1 - \lambda_2)}}\right)
$$

We also have,

$$
\frac{\eta_n}{2n} \sum_i U_{2,i} = O_P\left(\sqrt{\frac{\mathbb{E}\|Y_1\|^4}{n(\lambda_1 - \lambda_2)}}\right)
$$

Also note that while $U_{3,i}$ terms are 1-dependent, they are in fact uncorrelated. Thus, we have:

$$\text{Var}(\sum_i U_{3,i}) \leq \frac{M_d^2 n}{(\lambda_1 - \lambda_2)},$$

and,

$$\text{trace}(\mathbb{E}^* Z Z^T - \bar{\mathbb{V}}_n) = O_P\left(\sqrt{\frac{\mathbb{E}\|X_i X_i^T - \Sigma\|^4}{n(\lambda_1 - \lambda_2)}}\right)$$

Now we bound the Frobenius norm. We will start with the expected Frobenius norm of the first term of Eq S.26.

$$A_1 = \mathbb{E}\left\|\frac{1}{2n}\sum_{i=1}^n D_{i-1}Y_i v_1 v_1^T Y_i D_{i-1} - M_i\right\|_F^2$$

$$\leq \frac{1}{4n^2}\sum_i \mathbb{E}\|D_{i-1}Y_i v_1 v_1^T Y_i D_{i-1}\|_F^2 \leq \frac{E\|Y_1\|^4}{4n\eta_n(\lambda_1 - \lambda_2)}$$

Similarly,

$$A_2 = \mathbb{E}\left\|\frac{1}{n}\sum_i D_{i-1}Y_i v_1 v_1^T Y_{i-1}D_{i-1}\right\|_F^2$$

$$\leq \frac{1}{n\eta_n(\lambda_1 - \lambda_2)}M_d^2$$

Thus ,

$$\left\|\mathbb{E}^* Z Z^T - \bar{\mathbb{V}}_n\right\|_F = O_P\left(\sqrt{\frac{\mathbb{E}\|X_1 X_1^T - \Sigma\|^4}{n(\lambda_1 - \lambda_2)}}\right)$$

$\square$

## C.3 The Gaussian comparison lemma

We use the following lemma to compare to Gaussian random variables with mean 0 and different covariance matrices. Our result is related to [2], but our lemma below is easier to implement and does not require that $3\|\Sigma\|^2 \leq \|\Sigma\|_F^2$.

**Lemma C.1.** *[Comparison lemma for inner products of Gaussian random variables]*
*Suppose that $Z \sim N(0, \mathbb{V})$, $\check{Z} \sim N(0, \check{\mathbb{V}})$, $f = \|\mathbb{V}\|_F$, and $\Delta_1 = \text{tr}(\mathbb{V} - \check{\mathbb{V}})$. Then, there exists some constant $K > 0$ such that for any $\epsilon > 0$,*

$$\sup_t \left|P(Z^T Z \leq t) - P(\check{Z}^T \check{Z} \leq t)\right| \lesssim \sqrt{\frac{|\Delta_1| + \epsilon}{f}} + \exp\left\{-\left(\frac{\epsilon^2}{K^2\|\mathbb{V} - \check{\mathbb{V}}\|_F^2} \wedge \frac{\epsilon}{K\|\mathbb{V} - \check{\mathbb{V}}\|}\right)\right\}$$

*Proof.* Let $\lambda_1 \geq \ldots \geq \lambda_p$ denote the eigenvalues $\mathbb{V}$, $\gamma \geq \ldots \geq \gamma_p$ denote the eigenvalues of $\check{\mathbb{V}}$. Recall that $Z^T Z \sim \sum_{r=1}^p \lambda_r \eta_r$, $\check{Z}^T \check{Z} \sim \sum_{r=1}^p \gamma_r \eta_r$, where $\eta_r \sim \chi^2(1)$. It follows that:

$$P(Z^T Z \leq t) - P(\check{Z}^T \check{Z} \leq t)$$

$$= P\left(\frac{\sum_{r=1}^p \lambda_r \eta_r}{f} \leq \frac{t}{f}\right) - P\left(\frac{\sum_{r=1}^p \lambda_r \eta_r + \sum_{r=1}^p (\gamma_r - \lambda_r)\eta_r - \Delta_1}{f} \leq \frac{t - \Delta_1}{f}\right)$$

$$\leq P\left(\frac{t'}{f} \leq \frac{\sum_{r=1}^p \lambda_r \eta_r}{f} \leq \frac{t' + |\Delta_1| + \epsilon}{f}\right) + P\left(\left|\sum_{r=1}^p (\gamma_r - \lambda_r)\eta_r - \Delta_1\right| > \epsilon\right)$$

Observe that $\sum_{r=1}^p (\lambda_r - \gamma_r)^2 \leq \|\mathbb{V} - \check{\mathbb{V}}\|_F^2$ by Hoffman-Wielandt inequality and $\max_r |\lambda_r - \gamma_r| \leq \|\mathbb{V} - \check{\mathbb{V}}\|_{op}$ by Weyl's inequality. Since $\chi^2(1)$ is sub-Exponential, by Bernstein's inequality (see for example Theorem 2.8.2 of [7]:

$$P\left(\left|\sum_{r=1}^p (\gamma_r - \lambda_r)\eta_r - \Delta_1\right| > \epsilon\right) \leq \exp\left\{-\left(\frac{\epsilon^2}{K^2\|\mathbb{V} - \check{\mathbb{V}}\|_F^2} \wedge \frac{\epsilon}{K\|\mathbb{V} - \check{\mathbb{V}}\|}\right)\right\}$$

$\square$

## C.4   Other supporting lemmas for bootstrap consistency

Before presenting our supporting lemmas, we present some events we will use frequently. Let $\mathcal{A}_{\sin}$ denote the set

$$\mathcal{A}_{\sin} := \left\{1 - (v_1^T \hat{v}_1)^2 \leq \frac{\gamma_{\sin}}{\delta_{\sin}}\right\}. \tag{S.27}$$

Using Corollary 1, and the remark thereafter, we have:

$$P\left(1 - (v_1^T \hat{v}_1)^2 \geq \frac{\gamma_{\sin}}{\delta_{\sin}}\right) \leq \delta_{\sin}, \tag{S.28}$$

where, under the assumptions of Theorem 1,

$$\gamma_{\sin} = C_3 \frac{M_d \eta_n}{n(\lambda_1 - \lambda_2)} \tag{S.29}$$

Also let,

$$\mathcal{A}_n = \left\{\max_{1 \leq i \leq n} \|X_i\|_2^2 \leq \alpha_n\right\} \tag{S.30}$$

**Lemma C.2.** *[Bounding the norm of bootstrap residual from $T_1^*$] Let $\Delta_i = X_i X_i^T - X_{i-1} X_{i-1}^T$ and assume the conditions in Theorem 1. Let $D_i = V_\perp \Lambda_\perp^i V_\perp^T$, where $\Lambda_\perp(k, \ell) = \frac{1 + \eta_n \lambda_{k+1}/n}{1 + \eta_n \lambda_1/n} 1(k = \ell)$. For any $\epsilon, \delta > 0$, we have:*

$$P\left(\left\{\sqrt{\frac{n}{\eta_n}}\left\|\frac{\hat{V}_\perp \hat{V}_\perp^T T_1^* v_1 (v_1^T u_0)}{|v_1^T u_0|(1 + \eta_n \lambda_1/n)^n} - \frac{\eta_n}{n}\sum_i W_i D_{i-1} \Delta_i v_1\right\| \geq \epsilon\right\} \cap \mathcal{A}_n\right)$$

$$\leq C'' \frac{\alpha_n M_d \eta_n^3 \log d}{n \epsilon^2 \delta} + \delta$$

*Proof.*

$$\frac{\widehat{V}_\perp \widehat{V}_\perp^T T_1^* v_1 (v_1^T u_0)}{|v_1^T u_0|(1 + \eta_n \lambda_1/n)^{n-1}}$$

$$= \text{sign}(v_1^T u_0) \frac{\eta_n}{n} \sum_i W_i D_{i-1} \Delta_i v_1$$

$$+ \text{sign}(v_1^T u_0) \frac{\eta_n}{n} \underbrace{(\widehat{V}_\perp \widehat{V}_\perp^T - V_\perp V_\perp^T) \sum_i W_i D_{i-1} \Delta_i v_1}_{r_1}$$

$$+ \text{sign}(v_1^T u_0) \frac{\eta_n}{n} \left( \underbrace{\sum_i W_i \left( \frac{R_{1,i-1} \Delta_i v_1}{(1 + \lambda_1 \eta_n/n)^i} \right)}_{r_2} \right.$$

$$\left. + \underbrace{\frac{W_i (I + \eta_n \lambda_1/n)^{i-1} \Delta_i R_{i,n} v_1}{(1 + \lambda_1 \eta_n/n)^{n-1}}}_{r_3} + \underbrace{W_i \frac{R_{1,i-1} \Delta_i R_{i,n} v_1}{(1 + \lambda_1 \eta_n/n)^{n-1}}}_{r_4} \right)$$

Define

$$\mathcal{B}_{1,j} = \prod_{i=1}^{j} \left( I + \frac{\eta_n}{n} X_i X_i^T \right) \qquad \mathcal{B}_{j,n} = \prod_{i=j}^{n} \left( I + \frac{\eta_n}{n} X_i X_i^T \right) \tag{S.31}$$

When $j = 0$, $\mathcal{B}_{1,j} = I$.
Using Lemma B.1 we have:

$$R_{1,i} = \mathcal{B}_{1,i} - (I + \eta_n \Sigma/n)^i \qquad R_{i,n} = \mathcal{B}_{i,n} - (I + \eta_n \Sigma/n)^{n-i} \tag{S.32}$$

$$\mathbb{E}\|R_{1,i-1}\|^2 \le eM_d(1 + 2\log d)\frac{\eta_n^2}{n^2} i \left(1 + \eta_n \lambda_1/n\right)^{2i} \tag{S.33}$$

$$\mathbb{E}\|R_{i,n}\|^2 \le eM_d(1 + 2\log d)\frac{\eta_n^2}{n^2} (n - i) \left(1 + \eta_n \lambda_1/n\right)^{2(n-i)} \tag{S.34}$$

We have, on the good set $\mathcal{A}_{\text{sin}}$,

$$\mathbb{E}^*\|r_1\|^2 \le n\alpha_n \frac{\gamma_{\text{sin}}}{\delta_{\text{sin}}}$$

We also have:

$$\mathbb{E}\left[\mathbb{E}^*\|r_2\|^2 \mathbf{1}(\mathcal{A}_n)\right] \le \frac{\eta_n^2}{n^2} \alpha_n \sum_i \mathbb{E}[\|R_{1,i}^2 \mathbf{1}(\mathcal{A}_n)\|^2]$$

$$\le eM_d(1 + 2\log d)\alpha_n \eta_n^2$$

The last step is true because $\mathbb{E}[\|R_{1,i}^2 \mathbf{1}(\mathcal{A}_n)\|^2] \le \mathbb{E}[\|R_{1,i}^2\|^2]$. Similarly

$$\mathbb{E}\left[\mathbb{E}^*\|r_3\|^2 \mathbf{1}(\mathcal{A}_n)\right] \le eM_d(1 + 2\log d)\alpha_n \eta_n^2$$

and

$$\mathbb{E}\left[\mathbb{E}^*\|r_4\|^2 1(\mathcal{A}_n)\right] \leq e^2 M_d^2 (1 + 2\log d)^2 \alpha_n \eta_n^4/n$$

Finally, we have:

$$P\left(\left\{\frac{\eta_n}{n}\|\sum_j r_j\|^2 \geq \epsilon\right\} \cap \mathcal{A}_n\right) \leq P\left(\left\{4\frac{\eta_n}{n}\sum_j \|r_j\|^2 \geq \epsilon\right\} \cap \mathcal{A}_n\right)$$

$$\leq \sum_i P\left(\left\{\|r_i\|^2 \geq \frac{n\epsilon}{16\eta_n}\right\} \cap \mathcal{A}_n \cap \mathcal{A}_{\sin}\right) + \delta_{\sin}$$

$$\leq C \sum_i \mathbb{E}\left[\mathbb{E}^*\|r_i\|^2 1(\mathcal{A}_n \cap \mathcal{A}_{\sin})\right] \times \frac{\eta_n}{n\epsilon} + \delta_{\sin}$$

$$\overset{(i)}{\leq} C'\left(n\alpha_n \frac{\gamma_{\sin}}{\delta_{\sin}} + M_d \log d\alpha_n \eta_n^2\right) \times \frac{\eta_n}{n\epsilon} + \delta_{\sin}$$

$$\overset{(ii)}{\leq} C'' \frac{\alpha_n M_d \eta_n^3 \log d}{n\epsilon\delta_{\sin}} + \delta_{\sin}$$

Step (i) is true because $M_d \log d\eta_n^2/n \to 0$. Step (ii) is true because of Eq S.29. Now setting $\delta_{\sin}$ to any $\delta > 0$ gives the result.

$\square$

**Lemma C.3** (Concentration of the norm for the bootstrap)**.** *Let $u_0$ be uniformly distributed on $\mathbb{S}^{d-1}$ and $a_1 = u_0'v_1$ and $V_\perp V_\perp^T$ is orthogonal complement. Suppose that $(\alpha_n)_{n\geq 1}$ satisfies $0 \leq \frac{(\eta_n\alpha_n)^2}{n} \leq 1$. Then, for any $\epsilon > 0, 0 < \delta < 1$ and some $C > 0$,*

$$P\left(\left\{\left|\frac{\|B_n^*u_0\|}{|a_1|(1 + \eta_n\lambda_1/n)^n} - 1\right| \geq \epsilon_n\right\} \bigcap \mathcal{A}_n\right)$$

$$\leq \frac{d\exp\left(-\eta_n(\lambda_1 - \lambda_2) + \frac{\eta_n^2}{n}(\lambda_1^2 + M_d)\right) + \frac{\eta_n^2}{n}M_d\exp\left(\frac{\eta_n^2}{n}\right)}{8\log^{-1}(1/\delta)\delta^2 \ \epsilon^2 \left(1 + \frac{\eta_n^2\lambda_1^2}{n}\right)}$$

$$+ \frac{e^2\eta_n^2 M_d(1 + \log d)}{2n\epsilon^2} + \frac{C\beta_n^*\log(1/\delta)}{(1 - \beta_n^*)\delta^2\epsilon^2} + C\delta,$$

*where $\beta_n^*$ is defined in (S.36) and $\mathcal{A}_n$ is defined in Eq S.30.*

*Proof.* First note that we may reduce the problem as follows:

$$P\left(\left\{\left|\frac{\|B_n^*u_0\|}{|a_1|(1 + \eta_n\lambda_1/n)^n} - 1\right| \geq \epsilon\right\} \cap \mathcal{A}_n\right)$$

$$\leq P\left(\left\{\frac{\|B_n^*u_0 - B_n u_0\|_2}{|a_1|(1 + \eta_n\lambda_1/n)^n} + \left|\frac{\|B_n u_0\|_2}{|a_1|(1 + \eta_n\lambda_1/n)^n} - 1\right| > \epsilon\right\} \cap \mathcal{A}_n\right)$$

$$\leq \mathbb{E}\left[P^*\left(\frac{\|B_n^*u_0 - B_n u_0\|_2}{|a_1|(1 + \eta_n\lambda_1/n)^n} > \frac{\epsilon}{2}\right) 1(\mathcal{A}_n)\right] + P\left(\left|\frac{\|B_n u_0\|_2}{|a_1|(1 + \eta_n\lambda_1/n)^n} - 1\right| > \frac{\epsilon}{2}\right)$$

The bound for the second term follows from Lemma B.2. For the first term, we invoke Proposition B.2 so that, with probability at least $1 - C\delta$,

$$\frac{\|(B_n^* - B_n)g\|_2^2}{(v_1^T g)^2 (1 + \eta_n \lambda_1/n)^{2n}} \leq \frac{\log(1/\delta) \|B_n^* - B_n\|_F^2}{\delta^2 (1 + \eta_n \lambda_1/n)^{2n}}$$

Now, using the fact that for any two Hayek projections $P_S^*$ and $P_T^*$, $\mathbb{E}[(P_S^*)^T P_T^*] = 0$ and for any two matrices $\|AB\|_F \leq \|A\|_F \|B\|_{op}$, we have on the high probability set:

$$\mathbb{E}^* \|B_n^* - B_n\|_F^2$$
$$\leq \sum_{k=1}^{n} \sum_{|S|=k} \left(\frac{\eta_n}{n}\right)^{2k} \prod_{i=1}^{k} \left\|X_{S[i]} X'_{S[i]} - X_{S[i]-1} X'_{S[i]-1}\right\|_F^2 \prod_{j=1}^{k+1} \left\|\mathcal{B}_{j,n}^{(S)}\right\|_{op}^2,$$

where $\mathcal{B}_{j,n}^{(S)}$ denotes a contiguous block of $I + \frac{\eta_n}{n} X_i X_i^T$ only. More precisely, suppose $|S| = k$. Let $S[i]$ denote the $i$th element of $S$, with $S[0] = 0$ and $S[k+1] = n - 1$. For each $1 \leq j \leq k+1$ if $S[j] > S[j-1] + 1$ define $\mathcal{B}_{j,n}$ as:

$$\mathcal{B}_{j,n}^{(S)} = \prod_{i=S[j-1]+1}^{S[j]-1} \left(I + \frac{\eta_n}{n} X_i X_i^T\right) \tag{S.35}$$

otherwise, set $\mathcal{B}_{j,n}^{(S)} = I$. Now, we may repeat arguments in Lemma C.4 equations (S.37), (S.38), and (S.39) to conclude that, for some $C > 0$,

$$P\left(\frac{\log(1/\delta) \|B_n^* - B_n\|_F^2}{\delta^2 (1 + \eta_n \lambda_1/n)^{2n}} > \epsilon \bigcap \mathcal{A}_n\right) \leq \frac{C \log(1/\delta) \beta_n^*}{(1 - \beta_n^*) \delta^2 \epsilon^2}$$

The result follows. $\square$

**Lemma C.4** (Negligibility of higher-order Hoeffding projections for the bootstrap). *Suppose $\alpha_n$ is defined so that $0 \leq \beta_n^* \leq 1$, where*

$$\beta_n^* = \exp\left(\sqrt{\frac{CM_d^2 \eta_n^2 \log d}{n}}\right) \frac{4\eta_n^2 \alpha_n^2}{n} \tag{S.36}$$

*Then for any $\epsilon > 0, 0 < \delta < 1$ and for some $C > 0$,*

$$P\left(\left\{\frac{\sqrt{\frac{n}{\eta_n}} \left\|\hat{V}_\perp \hat{V}_\perp^T \sum_{k>1} T_k^* u_0\right\|}{|a_1|(1 + \frac{\eta_n \lambda_1}{n})^n} > \epsilon_n\right\} \bigcap \mathcal{A}_n\right)$$
$$\leq \exp\left(\sqrt{\frac{CM_d^2 \eta_n^2 \log d}{n}}\right) \frac{\log(1/\delta)}{\delta^2} \frac{\alpha_n^2 \beta_n^* \eta_n}{(1 - \beta_n^*) \epsilon^2} + C\delta,$$

*where $\mathcal{A}_n$ is defined in Eq S.30.*

*Proof.* Using the trace trick in Proposition B.2 again, we have that, with probability at least $1 - C\delta$ for some $C > 0$,

$$\frac{\frac{n}{\eta_n} \left\| \hat{V}_\perp \hat{V}_\perp^T \sum_{k>1} T_k^* g \right\|^2}{(v_1^T g)^2 (1 + \frac{\eta_n \lambda_1}{n})^{2n}} \leq \frac{\frac{n}{\eta_n} \log(1/\delta) \left\| \sum_{k>1} T_k \right\|_F^2}{\delta^2 (1 + \frac{\eta_n \lambda_1}{n})^{2n}}$$

The Hoeffding decomposition (Proposition A.4), together with the fact that $\|AB\|_F \leq \|A\|_F \|B\|_{op}$ implies:

$$\mathbb{E}^* \left[ \left\| \sum_{k>1} T_k^* \right\|_F^2 \right] = \mathbb{E}^* \left[ \sum_{k>1} \|T_k^*\|_F^2 \right]$$
$$\leq \sum_{k=2}^n \sum_{|S|=k} \left( \frac{\eta_n}{n} \right)^{2k} \prod_{i=1}^k \left\| X_{S[i]} X_{S[i]}^T - X_{S[i]-1} X_{S[i]-1}^T \right\|_F^2 \prod_{j=1}^{k+1} \left\| \mathcal{B}_{j,n}^{(S)} \right\|_{op}^2 \tag{S.37}$$

Now, that expectation corresponding to a given summand is given by:

$$\int_{\mathcal{A}_n} \left\| X_{S[i]} X_{S[i]}^T - X_{S[i]-1} X_{S[i]-1}^T \right\|_F^2 \prod_{j=1}^{k+1} \left\| \mathcal{B}_{j,n}^{(S)} \right\|^2 dP$$
$$\leq \int_{\mathcal{A}_n} \prod_{i=1}^k 4\alpha_n^2 \prod_{j=1}^{k+1} \left\| \mathcal{B}_{j,n}^{(S)} \right\|^2 dP \tag{S.38}$$
$$\leq (4\alpha_n^2)^k \prod_{j=1}^{k+1} \mathbb{E} \left[ \left\| \mathcal{B}_{j,n}^{(S)} \right\|^2 \right]$$

where $\mathcal{B}_{j,n}^{(S)}$ is defined in Eq S.35.

To bound $\mathbb{E} \left[ \left\| \mathcal{B}_{j,n}^{(S)} \right\|^2 \right]$, we invoke Lemma B.1 Eq S.21. For some $C > 0$ uniformly in $S$:

$$\prod_{j=1}^{k+1} \mathbb{E} \left[ \left\| \mathcal{B}_{j,n}^{(S)} \right\|^2 \right] \leq \exp \left( \sqrt{\frac{CM_d^2 \eta_n^2 \log d}{n}} \right)^{k+1} \left( 1 + \frac{\eta_n \lambda_1}{n} \right)^{2(n-k)}$$

Therefore, by Markov's inequality,

$$
P\left(\left\{\frac{\sqrt{\frac{n}{\eta_n}}\left\|\widehat{V}_\perp \widehat{V}_\perp^T \sum_{k>1} T_k^* u_0\right\|}{(1+\frac{\eta_n \lambda_1}{n})^n} > \epsilon_n\right\} \bigcap \mathcal{A}_n\right)
$$

$$
\leq \frac{n}{\delta^3 \epsilon_n^2 \eta_n} \exp\left(\sqrt{\frac{CM_d^2 \eta_n^2 \log d}{n}}\right) \sum_{k=2}^n \left(\frac{4\eta_n^2 \alpha_n^2}{n} \exp\left(\sqrt{\frac{CM_d^2 \eta_n^2 \log d}{n}}\right)\right)^k
$$

$$
\leq \alpha_n^2 \eta_n \delta_n^{-3} \epsilon_n^{-2} \exp\left(\sqrt{\frac{CM_d^2 \eta_n^2 \log d}{n}}\right) \sum_{k=1}^n \left(\frac{4\eta_n^2 \alpha_n^2}{n} \exp\left(\sqrt{\frac{CM_d^2 \eta_n^2 \log d}{n}}\right)\right)^k \tag{S.39}
$$

$$
\leq \exp\left(\sqrt{\frac{CM_d^2 \eta_n^2 \log d}{n}}\right) \frac{\alpha_n^2 \beta_n^* \eta_n}{(1-\beta_n^*)\epsilon_n^2 \delta_n^3}
$$

where the last line follows from a geometric series argument. $\qquad \square$

**Lemma C.5.**
$$
\sum_{i=0}^n \left(1 - \frac{\eta_n/n(\lambda_1-\lambda_2)}{1+\eta_n\lambda_1/n}\right)^{2i} \leq \frac{n}{\eta_n} \min\left(\eta_n, \frac{1}{\lambda_1-\lambda_2}\right)
$$

*Proof.* This follows from the definition of a geometric series. $\qquad \square$

**Lemma C.6** (Bounding the leading Hoeffding projection for the bootstrap on $V_\perp$)**.** *Let $\lambda_1 M_d (\log d)^2 \frac{\eta_n^2}{n} \to 0$, and $nd \exp(-\eta_n(\lambda_1-\lambda_2)) \to 0$. For any $\epsilon, \delta > 0$, and $C_1, C_2 \geq 0$, we have:*

$$
P\left(\left\{\sqrt{\frac{n}{\eta_n}}\frac{\|\widehat{V}_\perp \widehat{V}_\perp^T T_1^* V_\perp V_\perp^T u_0\|}{(1+\eta_n\lambda_1/n)^n |v_1^T u_0|} \geq \epsilon\right\} \cap \mathcal{A}_n\right) \leq \frac{C_1 \alpha_n M_d \eta_n^2 \log(1/\delta)}{n(\lambda_1-\lambda_2)\delta^3} \frac{1}{\epsilon^2} + C_2 \delta
$$

*Proof.* Using Proposition B.2, with probability at least $1-\delta$,

$$
\frac{\|\widehat{V}_\perp \widehat{V}_\perp^T T_1^* V_\perp V_\perp^T u_0\|^2}{(1+\eta_n\lambda_1/n)^{2n}\|v_1^T u_0\|^2} \leq \frac{\log(1/\delta)\left\|\widehat{V}_\perp \widehat{V}_\perp^T T_1^* V_\perp V_\perp^T\right\|_F^2}{\delta^2(1+\eta_n\lambda_1/n)^{2n}}
$$

$$
= \frac{\log(1/\delta)\text{trace}(\widehat{V}_\perp \widehat{V}_\perp^T T_1^* V_\perp V_\perp^T T_1^* \widehat{V}_\perp \widehat{V}_\perp^T)}{\delta^2(1+\eta_n\lambda_1/n)^{2n}}
$$

$$
= \frac{\log(1/\delta)\left\|\widehat{V}_\perp \widehat{V}_\perp^T T_1^* V_\perp\right\|_F^2}{\delta^2(1+\eta_n\lambda_1/n)^{2n}} \tag{S.40}
$$

First note that,

$$
\|V_\perp V_\perp^T - \widehat{V}_\perp \widehat{V}_\perp^T\|_F^2 = \|v_1 v_1^T - \hat{v}_1 \hat{v}_1^T\|_F^2 = 2(1-(v_1^T \hat{v}_1)^2)
$$

Thus, we have

$$
\mathbb{E}^* \|\widehat{V}_\perp \widehat{V}_\perp^T T_1^* V_\perp\|_F^2
$$

$$
= \frac{\eta_n^2}{n^2} \sum_i \|\widehat{V}_\perp \widehat{V}_\perp^T \mathcal{B}_{1,i-1}(X_i X_i^T - X_{i-1} X_{i-1}^T)\mathcal{B}_{i+1,n} V_\perp\|_F^2
$$

$$
\leq 4\frac{\eta_n^2}{n^2} \sum_i \sum_{j=1}^{6} \|r_{j,i}\|_F^2, \tag{S.41}
$$

where $B_{1,i}$ are defined in Eq S.32, and the residual vectors $r_{k,i}$ are defined as follows. Recall the definition of $R_{1,i}$ and $R_{i,n}$ from Eq S.32. Now define the following vectors which contribute to the remainder.

$$
r_{1,i} = \widehat{V}_\perp \widehat{V}_\perp^T R_{1,i-1}(Y_i - Y_{i-1})R_{i+1,n} V_\perp
$$

$$
r_{2,i} = \widehat{V}_\perp \widehat{V}_\perp^T R_{1,i-1}(Y_i - Y_{i-1})(I + \eta_n/n\Sigma)^{n-i} V_\perp
$$

$$
r_{3,i} = V_\perp V_\perp^T (I + \eta_n/n\Sigma)^{n-i}(Y_i - Y_{i-1})R_{i+1,n} V_\perp
$$

$$
r_{4,i} = V_\perp V_\perp^T (I + \eta_n/n\Sigma)^{n-i}(Y_i - Y_{i-1})(I + \eta_n/n\Sigma)^{n-i} V_\perp
$$

$$
r_{5,i} = (\widehat{V}_\perp \widehat{V}_\perp^T - V_\perp V_\perp^T)(I + \eta_n/n\Sigma)^{n-i}(Y_i - Y_{i-1})R_{i+1,n} V_\perp
$$

$$
r_{6,i} = (\widehat{V}_\perp \widehat{V}_\perp^T - V_\perp V_\perp^T)(I + \eta_n/n\Sigma)^{n-i}(Y_i - Y_{i-1})(I + \eta_n/n\Sigma)^{n-i} V_\perp
$$

First we will bound $\|r_{1,i}\|_F^2$. Recall the set $\mathcal{A}_n$ where the maximum norm is bounded from S.30.

$$
E_{1,i} := \int_{\mathcal{A}_n} \|r_{1,i}\|_F^2 dP \leq 2\alpha_n \int_{\mathcal{A}_n} \|R_{1,i-1}\|^2 \|R_{i+1,n}\|^2 dP
$$

$$
\leq 2\alpha_n \int \|R_{1,i}\|^2 \|R_{i+1,n}\|^2 dP \leq 2\alpha_n \mathbb{E}\|R_{1,i}\|^2 \mathbb{E}\|R_{i+1,n}\|^2 \tag{S.42}
$$

Similarly,

$$
E_{2,i} = \int_{\mathcal{A}_n} \|r_{2,i}\|_F^2 dP \leq 2\alpha_n \left(1 + \eta_n \lambda_2/n\right)^{2(n-i)} \mathbb{E}\|R_{1,i-1}\|^2 \tag{S.43}
$$

$$
E_{3,i} = \int_{\mathcal{A}_n} \|r_{3,i}\|_F^2 dP \leq 2\alpha_n \left(1 + \eta_n \lambda_2/n\right)^{2(i-1)} \mathbb{E}\|R_{i+1,n}\|^2 \tag{S.44}
$$

Similarly,

$$
E_{4,i} = \int_{\mathcal{A}_n} \|r_{4,i}\|_F^2 dP \leq 2\alpha_n \left(1 + \eta_n \lambda_2/n\right)^{2(n-1)} \tag{S.45}
$$

Recall the set $\mathcal{A}_{\sin}$ from Eq S.27. With probability at least $1 - \delta_{\sin}$,

$$
E_{5,i} = \int_{\mathcal{A}_n \cap \mathcal{A}_{\sin}} \|r_{5,i}\|_F^2 dP \leq 4\alpha_n \frac{\gamma_{\sin}}{\delta_{\sin}} \left(1 + \eta_n \lambda_1/n\right)^{2(i-1)} \mathbb{E}\|R_{i+1,n}\|^2
$$

$$E_{6,i} = \int_{\mathcal{A}_n \cap \mathcal{A}_{\sin}} \|r_{6,i}\|_F^2 dP \leq 2\alpha_n \frac{\gamma_{\sin}}{\delta_{\sin}} (1 + \eta_n\lambda_1/n)^{2(i-1)} (1 + \eta_n\lambda_2/n)^{2(n-i)}$$

Observe that, using Eq S.32, we have,

$$\mathcal{E}_1 := \sum_i E_{1,i} \leq \frac{2\alpha_n e^2 M_d^2 (1 + 2\log d)^2 \eta_n^4}{n}(1 + \eta_n\lambda_1/n)^{2(n-1)}$$

$$\mathcal{E}_2 := \sum_i (E_{2,i} + E_{3,i}) \leq \frac{4\alpha_n e M_d(1 + 2\log d)\eta_n^3}{n} \min\left(\eta_n, \frac{1}{\lambda_1 - \lambda_2}\right)(1 + \eta_n\lambda_1/n)^{2n-1}$$

$$\mathcal{E}_3 := \sum_i E_{4,i} \leq 2\alpha_n n (1 + \eta_n\lambda_2/n)^{2n}$$

With probability at least $1 - \delta_{\sin}$, we have

$$\mathcal{E}_4 := \sum_i E_{5,i} \leq 4\alpha_n \frac{\gamma_{\sin}}{\delta_{\sin}} e M_d(1 + 2\log d)\eta_n^2(1 + \eta_n\lambda_1)^{2(n-1)}$$

$$\mathcal{E}_5 := \sum_i E_{6,i} \leq 2\alpha_n \frac{\gamma_{\sin}}{\delta_{\sin}} \frac{n}{\eta_n} \min\left(\eta_n, \frac{1}{\lambda_1 - \lambda_2}\right)(1 + \eta_n\lambda_1)^{2(n-1)}$$

If $\lambda_1 M_d(\log d)^2 \frac{\eta_n^2}{n} \to 0$, then $\mathcal{E}_4 \leq C_1 \mathcal{E}_5$ for some positive constant $C_1$. If $nd\exp(-2\eta_n(\lambda_1 - \lambda_2)) \to 0$, then $\mathcal{E}_3 \leq C_2 \mathcal{E}_5$.

Thus, under these conditions,

$$\mathcal{E}_1, \mathcal{E}_2 \leq C_4 \mathcal{E}_5$$

With probability at least $1 - \delta_{\sin}$, for some positive constant $C'$,

$$\frac{\sum_{i=1}^5 \mathcal{E}_i}{(1 + \eta_n\lambda_1/n)^{2n}} \leq C'\alpha_n \frac{\gamma_{\sin}}{\delta_{\sin}}$$

Finally, using Eq S.41 we get:

$$\frac{\int_{\mathcal{A}_{\sin} \cap \mathcal{A}_n} \mathbb{E}^* \|\widehat{V}_\perp \widehat{V}_\perp^T T_1^* V_\perp\|_F^2 dP}{(1 + \eta_n\lambda_1/n)^{2n}} \leq C''\alpha_n \frac{\eta_n^2}{n} \frac{\gamma_{\sin}}{\delta_{\sin}} \tag{S.46}$$

Let $\mathcal{A}_1$ denote the set where Eq S.40 holds.

$$P\left(\left\{\frac{n}{\eta_n} \frac{\|\widehat{V}_\perp \widehat{V}_\perp^T T_1^* V_\perp V_\perp^T u_0\|^2}{(1 + \eta_n\lambda_1/n)^{2n}(v_1^T u_0)^2} \geq \epsilon\right\} \cap \mathcal{A}_n\right)$$

$$\leq P\left(\left\{\frac{\left\|\widehat{V}_\perp \widehat{V}_\perp^T T_1^* V_\perp\right\|_F^2}{(1 + \eta_n\lambda_1/n)^{2n}} \geq \frac{\epsilon\delta^2}{\log(1/\delta)} \frac{\eta_n}{n}\right\} \cap \mathcal{A}_n \cap \mathcal{A}_1\right) + 2\delta$$

$$\leq P\left(\left\{\frac{\left\|\widehat{V}_\perp \widehat{V}_\perp^T T_1^* V_\perp\right\|_F^2}{(1 + \eta_n\lambda_1/n)^{2n}} \geq \frac{\epsilon\delta^2}{\log(1/\delta)} \frac{\eta_n}{n}\right\} \cap \mathcal{A}_n \cap \mathcal{A}_1 \cap \mathcal{A}_{\sin}\right) + 2\delta + \delta_{\sin}$$

$$\leq \mathbb{E}\left[\frac{\mathbb{E}^*\left\|\widehat{V}_\perp \widehat{V}_\perp^T T_1^* V_\perp\right\|_F^2}{(1+\eta_n \lambda_1/n)^{2n}} \times \frac{\log(1/\delta)n}{\epsilon \delta^2 \eta_n} 1(\mathcal{A}_n \cap \mathcal{A}_1 \cap \mathcal{A}_{\sin})\right] + 2\delta + \delta_{\sin}$$

$$\overset{(i)}{\leq} \frac{C'' \alpha_n \eta_n \log(1/\delta)}{\delta_{\sin}\delta^2} \frac{\gamma_{\sin}}{\epsilon} + 2\delta + \delta_{\sin}$$

$$\overset{(ii)}{\leq} \frac{C''' \alpha_n M_d \eta_n^2 \log(1/\delta)}{n(\lambda_1 - \lambda_2)\delta_{\sin}\delta^2} \frac{1}{\epsilon} + 2\delta + \delta_{\sin}$$

Step (i) follows from Eq S.46. Step (ii) follows from the definition of $\gamma_{\sin}$ in Eq S.29. Now setting $\gamma_{\sin} = \delta$, we get the result. $\qquad\square$

# D   Proof of Proposition 1

*Proof of Proposition 1.* Since $\|X_{1j}\|_{\psi_2} \leq \nu_j$ it follows that $\left\|X_{1j}^2\right\|_{\psi_1} \leq \nu_j^2$. Observe that $(X_{1j}^2 - \mathbb{E}X_{1j}^2)/\nu_j^2$ is sub-Exponential with parameter at most 1 since $\left\|(X_{1j}^2 - \mathbb{E}[X_{1j}^2])/\nu_j^2\right\|_{\psi_1} \leq \left\|X_{1j}^2\right\|_{\psi_1}/\nu_j^2 = 1$. By multivariate Holder inequality with $p_j = \sum_{j=1}^d \nu_j^2/\nu_j^2$ and property (e) of Proposition 2.7.1 of [7], for $|\lambda| < 1/(\sum_{i=1}^d \nu_i^2)$:

$$\mathbb{E}\left[\exp\left(\lambda \sum_{j=1}^d (X_{1j}^2 - \mathbb{E}[X_{1j}^2])\right)\right] \leq \prod_{j=1}^d \mathbb{E}\left[\exp\left(\lambda(X_{1j}^2 - \mathbb{E}[X_{1j}^2])\right)^{\frac{\sum_{i=1}^d \nu_i^2}{\nu_j^2}}\right]^{\frac{\nu_j^2}{\sum_{i=1}^d \nu_i^2}}$$

$$= \prod_{j=1}^d \mathbb{E}\left[\exp\left(\frac{\lambda(\sum_{i=1}^d \nu_i^2)(X_{1j}^2 - \mathbb{E}[X_{1j}^2])}{\nu_j^2}\right)\right]^{\frac{\nu_j^2}{\sum_{i=1}^d \nu_i^2}}$$

$$\leq \prod_{j=1}^d \exp\left(\frac{K\lambda^2(\sum_{i=1}^d \nu_i^2)^2 \nu_j^2}{\sum_{i=1}^d \nu_i^2}\right)$$

$$= \exp\left\{K\lambda^2 \left(\sum_{i=1}^d \nu_i^2\right)^2\right\}$$

Therefore, $\left\|\sum_{i=1}^d X_{1i}^2\right\|_{\psi_1} \leq \sum_{i=1}^d \nu_i^2$. Since a subexponential random variable $T$ satisfy the tail condition:

$$P(T - \mathbb{E}[T] > t) \leq \exp(-t/K\nu)$$

for another universal constant $K > 0$, the second claim follows by a union bound and noting that $\mathbb{E}[\|X_1\|_2^2] \leq \sum_{i=1}^d \nu_i^2 < C_2$ since absolute summability implies square summability. $\qquad\square$