# OpenReview forum: "Bootstrapping the Error of Oja's Algorithm "
_NeurIPS.cc/2021/Conference — NeurIPS 2021 Spotlight_

### Official Review · Reviewer_tCtM · 2021-07-06

**Rating:** 8
**Confidence:** 3

**Summary:**

Oja’s principal subspace algorithm is a well-known and powerful technique for learning and tracking principal information in a time series setting. In this paper, the authors use U-statistics and a concentration of matrix products to obtain an error estimate between the leading eigenvector and the output of Oja’s algorithm. The other eigenvectors are probably beyond the theoretical understanding of random matrix theory at the moment. There are two main contributions: (1) Establishes the distribution of the sin^2 error estimate for the leading eigenvector computed via Oja’s algorithm and (2) A stream-friendly bootstrap algorithm to compute quantiles of the error estimate. The paper contains an interesting theoretical analysis involving modern matrix random theory, and since they have the distribution, a potential application to bootstrapping. The main novelties in this paper are theoretical, and the author could better demonstrate the advantages of their online multiplier bootstrapping technique.

**Limitations And Societal Impact:**

The expected limitations were discussed such as the analysis is only for the first principal vector of Oja's algorithm.

**Main Review:**

The paper's main contribution is theoretical, and so to truly appreciate this paper, one must get into the technical details. The authors take the standard point-of-view and analyze Oja's algorithm as a limit of products of random variables. I am very impressed that they were able to establish the distribution of the sin^2 error between the leading eigenvector of X and the computed vector via Oja’s algorithm (see Theorem 1). It is implicit as the probability distribution is connected to the eigenvalues and eigenvectors of Sigma, limiting its practical application. To have some practice application of Theorem 1, the authors propose a streaming bootstrap procedure (see Theorem 2).

The paper explains an overview of their analysis. After n iterations, Oja's algorithm can be expressed as a normalized matrix-vector product with a matrix B_n. Using Hoeffding decomposition, the authors write B_n as a sum of matrices that are uncorrelated (see (5)). This allows them to compute second-moment information of norms of B_n and consequential the distribution of the sin^2 error. For the bootstrapping idea, the authors use a comparison lemma.

While I don't fully appreciate all the 36 pages of technical details in the supplementary materials, I hope that the authors can better describe their final practical algorithm. In fact, I have several confusions involving Algorithm 1:

The input is u_0, but at the same time u_0 is assigned g/||g||. I think the authors want u_0 to be uniform on the sphere, so it shouldn't be an input. Also, what is Z_i in the algorithm (how to estimate their distribution)? I encourage the author's to write "Compute Y_t = X_tX_t^T - \Sigma" in the algorithm? How does one compute Sigma in Algorithm 1? What is the integer p in Algorithm 1? Theorem 1 needs the stepsize to go to infinite, but in Algorithm 1, it is constant. While Algorithm 1 presents the stepsize as constant, Section 4 uses log(n) (but what happens at n = 1?). Also, Theorem 1 presents an asymptotic result, but do the authors understand the optimal step size for the fastest convergence in probability?

Here are a few grammar errors that I spotted while reading. I am sure the authors will want to correct them.

l30: Define X.

l65: "convergence rates which deteriorate" should read "convergence rates that deteriorate"

l105: "where the the random" should read "where the random"

l161: "approximation to the distribution" should read "approximation of the distribution"

l163: "wew update" should read "we update"

l189: "growly wih n" should be "grow with n"

l191: "multivariate Gaussian distribution", do the authors mean "multivariate standard Gaussian distribution"? Otherwise, is the statement true?

l249: "parameters of this distributions" should read "parameters of this distribution"

l260: "we use the later" should read "we use the latter"

l263: "that that of the" should read "that of the"

l276: "For step 2-3" should read "For steps 2-3"

References: put captials for: oja, bernstein, pca, fpca, robbins-monro, etc.

**Time Spent Reviewing:**

6

---

> ### Author Response · Authors · 2021-08-10
> **Response to Reviewer tCtM**
>
> 1. We thank the reviewer for the encouraging remarks.  The reviewer is correct in that the central limit theorem,while important, is not directly applicable to inference since the variance depends on the eigenvalues and eigenvectors of $\Sigma$. Our bootstrap mimics the distribution of the $\sin^2$ error, bypassing direct estimation.  We would also like to point out that our procedure also avoids estimating the covariance matrix, which is needed with a more typical multiplier bootstrap procedure.
> 2. We will remove $u_0$ as an input to the algorithm.
> 3. Regarding Algorithm 1: We apologize for the typo.  $Z_i$ should be $W_i$. $p$ should be $d$. $Y_t$ is also a typo and should be $h^{(i)}$. The traditional Oja update is $v \leftarrow (I+\eta X_tX_t^T)v$. However, for the $i^{th}$ bootstrap replicate, we use $$v^{(i)}\leftarrow(I+\eta X_tX_t^T+\eta W_i(X_tX_t^T-X_{t-1}X_{t-1}^T))v^{(i)}, \ \ \ \text{where  } \eta=\eta_n/n.$$
> 4. Regarding constant step-size in Algorithm 1 - For a size  $n$ dataset we use a step-size $\eta$ which is dependent on $n$, but does not change with $t$, the number of data-points seen so far.  Therefore, it is appropriate to view the step size as a fixed input for a given sample size. Also note that our step size is parametrized as $\eta=\eta_n/n$, where $\eta_n$ has to go to infinity. A more detailed comment on fixed step-sizes is provided in point 7 of our response to reviewer Kj1d.
> 5.Regarding the optimal step-size - it should be noted that the primary goal of our paper is to develop confidence intervals for the $\sin^2$ error that hold over a range of tuning parameters rather than deriving optimal rates of convergence.  Nevertheless, as a byproduct of our analysis, we see that the convergence rate is $\eta_n/n$; therefore to choose the optimal step size, we need to make $\eta_n$ as small as possible while still satisfying the condition $nd \exp( - \eta_n ( \lambda_1 - \lambda_2)) \rightarrow 0$. This basically gives us comparable error rates as in [28] up-to logarithmic factors (see Remark 2).
> 6. We thank the reviewer for pointing out the typographical errors -- we will correct them.  We also plan to provide additional explanation of Algorithm 1.

---

> > ### Comment · Reviewer_tCtM · 2021-08-29
> > **Reply**
> >
> > Thank you for your careful response. A very welcomed paper in my opinion. My rating stays the same.

---

### Official Review · Reviewer_Kj1d · 2021-07-17

**Rating:** 6
**Confidence:** 4

**Summary:**

In this paper the bootstrapped Oja scheme is considered. For the squared sin distance between the Oja’s scheme output and the first eigenvector of the covariance matrix of observations, the authors provide a Gaussian approximation and demonstrate bootstrap consistency.

**Limitations And Societal Impact:**

The learning rate in the considered version of Oja scheme assumes the knowledge of the number of upcoming observations (e.g. see [28]).
Would not it be a limitation for the streaming PCA problem?

**Main Review:**

The authors consider a problem of quantification of uncertainty for Oja scheme in the framework for the streaming PCA. With this aim, the multiplier bootstrap is introduced in the update step of the Oja scheme. The proof of asymptotical Gaussian approximation for the squared sin distance and the asymptotical first term expansion of the squared sin error is based on the Hoeffding decomposition of the update matrices and high-dimensional CLT for the quadratic forms of random vectors and concentration and anti-concentration bounds. For the bootstrapped Oja the standard techniques lead to the bootstrap consistency result. In the end, the authors demonstrate the convergence of the bootstrapped CDF to the sampling CDF for the case of well-posed problems and further discuss the limitations.

The paper is overall quite well written. The following points could be corrected and clarified:
1) S and S_k in the Eq. 5 are not defined
2) In Algorithm 1 in the update step a vector estimate is missing in the brackets
3) In Algorithm 1 Z_i -> W_i
4) In Remark 2, which is the referenced result? In [28] the error has log (d) in the nominator, and in the Remark 2 it is log(n)
5) I am not sure I understand the condition (10) in view of (11), does the left-hand side of (10) contain anything depending on n?
6) In supplementary material in Proposition A2 a number of the equation is missing


**Time Spent Reviewing:**

5

---

> ### Author Response · Authors · 2021-08-10
> **Response to Reviewer Kj1d**
>
> 1. We want to respectfully point out that our proof techniques for establishing bootstrap consistency are far from standard.  For the bootstrap, we have a product of weakly dependent matrices arising out of the update rule, which is in many ways more difficult to analyze than the original Oja vector.  Applying the Gaussian Comparison Lemma, which we also had to establish for our problem, is also nonstandard; see the proof of Lemma B.1 and Lemma 1.  Finally, a naive analysis of the remainder yields highly sub-optimal dependence on $d$, which is why one needs to be careful about bounding the remainder terms.
> 2. Regarding Eq.5 - $\mathcal{S}_k$ in Eq. 5 is all subsets of size $k$ and $S$ is a subset of this set of subsets.
> 3. Regarding Algorithm 1 - We apologize for the typo.  $Z_i$ should be $W_i$.  $t$ denotes the index of streaming data points. $i$ denotes the index of the bootstrap replicate. $Y_t$ is also a typo and should be $h^{(i)}$. The traditional Oja update is $v \leftarrow (I+\eta X_tX_t^T)v$. However, for the $i^{th}$ bootstrap replicate, we use $$v^{(i)}\leftarrow(I+\eta X_tX_t^T+\eta W_i(X_tX_t^T-X_{t-1}X_{t-1}^T))v^{(i)}, \ \ \ \text{where  } \eta=\eta_n/n.$$
> 4. Regarding Remark 2 - While our main goal in Remark 2 was to show the error rates are similar to existing work up-to logarithmic factors, we will slightly modify the step size so that $\eta_n = \frac{\log (d \vee n)}{n(\lambda_1 - \lambda_2)}$, thus satisfying the condition $nd \exp( - \eta_n ( \lambda_1 - \lambda_2)) \rightarrow 0$. This is very similar to what was derived in [28].
> 5. Regarding the condition in Eq (10) - We are considering a high-dimensional triangular array setup, where $d$ and other parameters depend on the sample size $n$.  We will clarify this further.
> 6. We will fix the typographical errors.
> 7.Regarding a fixed learning rate that requires knowledge of $n$ - First, note that even with the simplification of a fixed learning rate, the analysis is novel and nontrivial. For the case of a fixed rate, note that when $M_d$ and $\lambda_1-\lambda_2$ are constants and $d$ is polynomial (this is for ease of explanation; certain exponential regimes also work) , $\eta_n$ can be any number between $\log (n\vee d)$ and $n^{1/3 -\delta}$ for some small $\delta >0$. Since this is a wide range, guessing $n$ incorrectly will not cause problems so long as the guess is not off by many orders of magnitude.  Furthermore, with the adaptive learning rate, a subtle issue arises with the Lindeberg condition in the Central Limit Theorem even for the simpler fixed-dimension case. To be concrete, if we use a learning rate proportional to $1/t$, then the Hajek projection ends up being a sum of independent random variables where the first few observations have much more weight than the later ones. For this adaptive regime, the Lindeberg condition is not satisfied. However, it is entirely possible to come up with other adaptive schemes that do not require the knowledge of $n$ and are likely to satisfy the Lindeberg condition. However, given how involved the proof is with known $n$, we leave this for future work.

---

> > ### Comment · Reviewer_Kj1d · 2021-08-31
> > **Reply**
> >
> > Thank you very much for your response. I will leave the score unchanged.

---

### Official Review · Reviewer_H32z · 2021-07-18

**Rating:** 6
**Confidence:** 3

**Summary:**

This paper study the problem of uncertainty quantification for the estimation error of the leading eigenvector from Oja's algorithm. The authors establish the distribution of $\sin^2$ error of the leading eigenvetor of Oja's algorithm. Also, the authors propose an online bootstrap algorithm for Oja's algorithm and provide the conditions under which the bootstrap is consistent.


**Limitations And Societal Impact:**

See main review

**Main Review:**


Strength:
1. Bootstrapping the error of Oja's algorithm is an interesting and important problem. The paper provides solid contributions to this problem.
2. The distribution of $\sin^2$ error of the leading eigenvetor of Oja's algorithm is novel.


Weakness:
1. The writing is not good. Some notations are confusing. For example, the variables $Y_t$ and $Z_i$ in Algorithm 1 are not defined. Also, some expectation notations lack brackets. For instance, in line 259, $\mathbb{E}ZZ^T$ should be $\mathbb{E}[ZZ^T]$.
2. The second comment is still about the writing. In Figure 1, what does the x-axis represent? Why the range of the x-axis of figures (C) and (D) are [0,1]? I suggest the authors can provide more descriptions on the figures.
3. I'm confused about the data generation of the experiment. The paper chooses $X=Z(\Sigma^{1/2})^T$, which indicates that $\mathbb{E}[X^T X]=\Sigma^{1/2}\mathbb{E}[Z^T Z]\Sigma^{1/2}$. Since $\mathbb{E}[Z^T Z]=nI$, the expectation of$X^T X$ should be $n\Sigma$ rather than $\Sigma$. Would you please tell me if I have missed something?

Overall, I think the paper provides some solid contributions to an important problem, but the writing of the paper should be polished.

**Time Spent Reviewing:**

6

---

> ### Author Response · Authors · 2021-08-10
> **Response to Reviewer H32z**
>
> 1. We thank the reviewer for their kind words regarding the importance of the problem and the novelty of our analysis.
> 2. Regarding Algorithm 1 -  We apologize for the typo.  $Z_i$ should be $W_i$.  $t$ denotes the index of streaming data points. $i$ denotes the index of the bootstrap replicate. $Y_t$ is also a typo and should be $h^{(i)}$. The traditional Oja update is $v = (I+\eta X_tX_t^T)v$. However, for the $i^{th}$ bootstrap replicate, we use $$v^{(i)}\leftarrow(I+\eta X_tX_t^T+\eta W_i(X_tX_t^T-X_{t-1}X_{t-1}^T))v^{(i)}, \ \ \ \text{where  } \eta=\eta_n/n.$$
> 3. Regarding the X axis - the X axis is simply the values of the $sin^2$ error times $1/\eta$ where $\eta = \log n/n$. For (A) and (B) the variance decay does not satisfy our theorem’s conditions and thus, the normalized error does not behave like a $O_P(1)$ random variable. However, for (C) and (D) the variance decay satisfies the conditions and in this case the normalized error is $O_P(1)$, which happens to be in the [0,1] range for this example. We will explain this clearly.
> 4. Confusion about the data generating process - you are quite right. We will correct this typo, $E[X^TX]$ is indeed $n\Sigma$.

---

> > ### Comment · Reviewer_H32z · 2021-08-30
> > **Reply**
> >
> > Thanks to the authors for the reply. I will leave my score unchanged. I suggest authors fixing typos and incorporating the explanation to the revision.

---

### Official Review · Reviewer_2S7D · 2021-07-19

**Rating:** 6
**Confidence:** 3

**Summary:**

This paper develops an online bootstrap method to quantify the error of estimating the leading eigenvector of a matrix via Oja's algorithm. The authors a) find a closed from expression of the leading eigenvector from Oja's algorithm, b) approximate this closed from expression via Hoeffding decomposition for U-statistics, and c) devise a Gaussian multiplier bootstrap (algorithm) to approximate the distribution of the Hoeffding decomposition. The authors establish Gaussian approximation and bootstrap consistency results and include a small scale simulation study.

**Limitations And Societal Impact:**

I wished the authors had discussed how to choose the tuning parameters of their algorithm and included a more comprehensive simulation study. (The current simulation is based on a covariance matrix with very light correlation/ almost sparse. How does their method work on dense and/ or equicorrelated covariances)?

**Main Review:**

This is an interesting and well-written paper which balances rigorous theory and intuition in just the right way. I particularly enjoyed how the authors draw from classical statistical theory (U-statistics, Hoeffding decomposiiton) and modern theory (Gaussian approximation, hiogh-dim. bootstrap) to solve a relevant problem in machine learning.

I have the following comments:
1) Unfortunately, I do not understand Algorithm 1: Where are the $W_i$'s used? What are $Y_t$ and $Z_i$? I would appreciate if the authors could explain and clarify the notation.
2) Reg Theorem 1: To choose $\eta_n$ one has to gauge $\lambda_1 - \lambda_2$ and $M_d$. How do the authors do this in practice? How sensitive is their algorithm to values of $\eta_n$?
3) Line 187: Technically, convergence of the Kolmogorov distance in eq. (11) does not imply that the authors "establish the [limiting] distribution of $n/\eta_n(1- (v_1^T\hat{v}_1)^2)$" because the limiting distribution of $Z_n^TZ_n$ need not to exist. It would be more accurate to state that in the limit the difference of the distributions of $n/\eta_n(1- (v_1^T\hat{v}_1)^2)$ and $Z_n^TZ_n$ in Kolmogorov distance vanishes. And that's good enough for most (all?) statistical purposes.

**Time Spent Reviewing:**

1.5

---

> ### Author Response · Authors · 2021-08-10
> **Response to Reviewer 2S7D**
>
> 1. Thank you for your encouraging words. Our paper brings together ideas from classical and modern statistical theory as well as recent results from random matrices to justify an inference procedure for streaming PCA. To our knowledge, we are the first to obtain a distributional approximation and consistency of a novel online bootstrap procedure for this problem. We believe that our framework will apply to many other streaming inference/non-convex optimization problems that use a similar update rule.
> 2. Regarding Algorithm 1 - We apologize for the typo.  $Z_i$ should be $W_i$.  $t$ denotes the index of streaming data points. $i$ denotes the index of the bootstrap replicate. $Y_t$ is also a typo and should be $h^{(i)}$. The traditional Oja update is $v \leftarrow (I+\eta X_tX_t^T)v$. However, for the $i^{th}$ bootstrap replicate, we use: $$v^{(i)}\leftarrow(I+\eta X_tX_t^T+\eta W_i(X_tX_t^T-X_{t-1}X_{t-1}^T))v^{(i)}, \ \ \ \text{where  } \eta=\eta_n/n.$$
> 3. Regarding Theorem 1 - It is true that the optimal choice of $\eta_n$, which can be ascertained from the normalizing factor in the Central Limit Theorem, depends on both $\lambda_1 - \lambda_2$ and $M_d$.  We first want to point out that almost all theoretical analyses of streaming PCA have dependence on the eigengap and norms of $X_iX_i^T-\Sigma$ and their functions. We are not trying to come up with a better analysis of streaming PCA. We are providing the first online bootstrap algorithm with provable guarantees in the high dimensional setting. Our central limit theorem and bootstrap results hold over a wide range of tuning parameters.  Our primary goal is to construct confidence intervals for the $\sin^2$ error given $\eta_n$.
> 4. Regarding line 187 -  You are correct. We had removed all but that reference to weak convergence. We will remove that in the future version.
> 5. Limitations about experiments with light correlation -  Note that the correlation structure of the covariance matrix is immaterial as long as there is sufficient variance decay. Our experiments show that when variance decay is reasonably fast bootstrap provides a good approximation of the sampling distribution. However, we did experiments with an equi-correlation matrix where all off diagonal entries of the correlation matrix are set to a fixed number which we varied between 0.05 to 0.5. As long as the variance decay is present, the qualitative trends are the same, as expected. We also want to point out that the experiments are simply there as proof of concept, since our goal is not to present a new and improved algorithm or analysis of the streaming PCA problem.

---

> > ### Comment · Reviewer_2S7D · 2021-09-18
> > **Reply**
> >
> > I'd like to thank the authors for their reply. I suggest authors fixing typos and incorporating the explanation to the revision .I will not change my score and am very happy with the paper.

---

### Decision · Program_Chairs · 2021-09-27

**Decision:**

Accept (Spotlight)

**Comment:**

The reviewers were unanimous in their support for this paper, with a clear theoretical contribution to our understanding of an important algorithm. Most reviewers also felt the paper was well-written, and we thus feel it will be broadly interesting to the NeurIPS community.